# Development of a Two-Dimensional Hybrid Sediment-Transport Model

Yaoxin Zhang [1,*] , Mohammad Al-Hamdan [1,2,3] and Daniel Wren [4]

1 National Center for Computational Hydroscience and Engineering, University of Mississippi, Oxford, MS 38655, USA; yzhang@olemiss.edu
2 Department of Civil Engineering, University of Mississippi, University, MS 38677, USA; mzalhamd@olemiss.edu
3 Department of Geology and Geological Engineering, University of Mississippi, University, MS 38677, USA
4 National Sedimentation Laboratory (NSL), USDA Agricultural Research Service (ARS), Oxford, MS 38655, USA; daniel.wren@usda.gov
* Correspondence: yzhang@ncche.olemiss.edu or yaoxin@olemiss.edu; Tel.: +1-6629158972

**Abstract:** This paper presents the development of a two-dimensional hydrodynamic sediment transport model using the finite volume method based on a collocated unstructured hybrid-mesh system consisting of triangular and quadrilateral cells. The model is a single-phase nonequilibrium sediment-transport model for nonuniform and noncohesive sediments in unsteady turbulent flows that considers multiple sediment-transport processes such as deposition, erosion, transport, and bed sorting. This model features a hybrid unstructured mesh system for easy mesh generation in complex domains. To avoid interpolation from vertices in conventional unstructured models, this model adopted a second-order accurate edge-gradient evaluation method to consider the mesh irregularities based on Taylor's series expansion. In addition, the multipoint momentum interpolation corrections were integrated to avoid possible nonphysical oscillations during the wetting-and-drying process, common in unsteady sediment transport problems, to ensure both numerical stability and numerical accuracy. The developed sediment transport model was validated by a benchmark degradation case for the erosion process with armoring effects, a benchmark aggradation case for the deposition process, and a naturally meandering river for long-term unsteady sediment-transport processes. Finally, the model was successfully applied to simulate sediment transport in a reservoir that was significantly affected by typhoon events.

**Keywords:** deposition; erosion; unsteady; nonequilibrium





## 1. Introduction

Sediment transport, defined as sediments driven by water and moving with the water, is one of the most important processes when studying morphological and environmental problems. Sediment transport may result in sediment depositions in lakes, reservoirs [1,2] and coastal wetlands [3,4]; erosion of riverbanks [5,6], coastlines [7,8], and downstream of dams [9,10]; local scour downstream of hydraulic structures [11,12]; gully erosion [13,14]; channel evolution [15,16]; adsorption/de-adsorption and resuspension [17], etc.

In addition to conventional physical models, with the advancement of computer technology and numerical methods, numerical models have become powerful tools to study sediment transport in rivers, lakes, reservoirs, and coastal regions. In natural rivers, sediment in the water can be described as a two-phase flow [18,19], in which the sediment phase and the water phase are governed by the continuity and the momentum equations and interact with each other all the time [20,21]. In the conventional single-phase flow method, sediment transport is described as a diffusion phenomenon using the momentum equation and the transport equations: the suspended load moves in the form of a suspension in the water column, while the bed load moves by sliding, rolling, or saltation along or near

the bed. Despite the advancement of the two-phase flow method [21–23] in recent years, the single-phase flow method [20,24–26] is still dominant in sediment transport modeling due to its relative simplicity and higher computational efficiency.

Sediment transport models may be categorized according to the dimension of study domains (one dimensional (1D), two dimensional (2D), or three dimensional (3D)), flow conditions (steady or unsteady), uniformity of sediment particles (single-sized or multiple-sized), transport mode of sediment (equilibrium or nonequilibrium), and the cohesion of sediment materials (noncohesive or cohesive). In natural rivers, nonuniform and noncohesive sediments are dominant, and the flows are unsteady and turbulent. Moving sediments interact with the channel and can both change channel geometry and be influenced by channel geometry. The dynamic interactions between flow and sediments imply that global equilibrium sediment transport (the sediment transport rate equals the transport capacity) is rare. Local transport may be temporarily in equilibrium with the flow, however, nonequilibrium sediment transport is much more common in natural channels [18]. The 1D sediment transport models (e.g., [25]; CCHE1D [27]; MIKE 11 [28]; HEC-RAS [29]; and GSTAR-1D [30]) were developed for averaged solutions with high computing efficiency for long-term simulations in large-scaled domains, while 3D models (e.g., CCHE3D [31]; Delft3D [32]) are for highly-sophisticated solutions of short-term simulations in local small domains. The 2D models (e.g., [19]; CCHE2D [24]; SRH-2D [26]; [33]) are in between 1D and 3D models with respect to computing efficiency and numerical accuracy. Theoretically, sediment transport is a 3D phenomenon and 3D models should be used. However, in practice for different problems, model selection needs to balance computing efficiency and numerical accuracy.

Unlike hydrodynamic flow models, there are more uncertainties and challenges with the sediment transport models since numerous semiempirical and empirical formulas have been developed, covering all aspects of the sediment transport process, such as the sediment settling velocity, sediment incipient motion, mobile bed roughness, critical shear stress, suspended-load transport capacity, and bed-load transport capacity. A good review of some of those formulas can be found in [20]. Most of those formulas were derived based on physical laws using experimental data under steady flow conditions, and application of them to truly unsteady sediment transport has been a concern [21]. Each formula has its own advantages, limitations, and application ranges. A sediment transport model cannot accommodate all the available formulas, which would make the model difficult to use. The same model cannot use multiple approaches in most cases since different formulas often yield significantly different simulation results according to the authors' experiences.

In the review of sediment transport models by Papanicolaou et al. [34], model limitations were identified, including turbulence calculations, entrainment formulas established under uniform flow conditions, fractional (nonuniform) sediment transport calculations, evaluation of sediment dispersion and diffusion coefficient, soil contributions from banks, hill slopes, and floodplains, and two-phase flow modeling. Most of these problems have not been resolved. Basically, sediment transport modeling needs to address the unsteady and nonequilibrium natures of sediment transport [26]. On the other hand, the mesh system (structured or unstructured) determines the numerical methods in sediment transport models. Despite their advantages and disadvantages, both structured and unstructured meshes have been widely used. For example, the CCHE2D model [19,24] and the Delta3D model [32] were based on structured meshes, while SRH-2D [26] was based on unstructured polygon meshes. Compared to structured meshes, unstructured meshes are more suitable and flexible for geometrically complex domains with complicated boundaries. Among unstructured meshes, triangle meshes, and quadrilateral meshes are the most popular and easiest to generate [35]. In addition, structured (quadrilateral) meshes can be shared by both structured models and unstructured models in many cases.

In this study, an alternative single-phase 2D depth-integrated sediment transport model for turbulent flows was developed. The model aims to simulate unsteady, nonequilibrium, and nonuniform (fractional) sediment transport processes for noncohesive sedi-



ment materials on both the laboratory scale and the field scale. The model features (1) an unstructured hybrid-mesh system consisting of either triangle cells or quadrilateral cells or mixed triangle and quadrilateral cells for geometrically-complex domains with high adaptivity; (2) an edge gradient evaluation method with second-order accuracy designed for mesh irregularity and nonuniformity; and, (3) a multipoint momentum interpolation correction method to remove possible nonphysical oscillations in the wetting-and-drying process. The development details are presented in the Section 2 (Numerical Model) and Section 3 (Numerical Method). Selected examples and applications will demonstrate and validate the proposed sediment-transport model.

## 2. Numerical Model

### 2.1. Flow Model

The flow model is the backbone of the sediment-transport model. The proposed sediment-transport model is based on a 2D depth-integrated hydrodynamic-flow model developed by Zhang et al. [36] where the continuity equation and the momentum equations for unsteady turbulent flow are:

$$\frac{\partial h}{\partial t} + \frac{\partial (hu)}{\partial x} + \frac{\partial (hv)}{\partial y} = 0 \tag{1}$$

$$\frac{\partial (hu)}{\partial t} + \frac{\partial (hu^2)}{\partial x} + \frac{\partial (huv)}{\partial y} = -gh\frac{\partial \eta}{\partial x} + \frac{1}{\rho}\left[\frac{\partial (h\tau_{xx})}{\partial x} + \frac{\partial (h\tau_{xy})}{\partial y}\right] + \frac{(\tau_{wx} - \tau_{bx})}{\rho} + f_{cor}hv \tag{2}$$

$$\frac{\partial (hv)}{\partial t} + \frac{\partial (huv)}{\partial x} + \frac{\partial (hv^2)}{\partial y} = -gh\frac{\partial \eta}{\partial y} + \frac{1}{\rho}\left[\frac{\partial (h\tau_{yx})}{\partial x} + \frac{\partial (h\tau_{yy})}{\partial y}\right] + \frac{(\tau_{wy} - \tau_{by})}{\rho} - f_{cor}hu \tag{3}$$

where $t$ represents time (s); $u$ and $v$ are depth-integrated velocity (m/s) components in the $x$ and $y$ directions, respectively; $\eta$ is the water surface elevation (m); $\rho$ is the water density (kg/m$^3$); $h = \eta - z_b$ is the local water depth (m) and $z_b$ is the bed elevation (m); $g$ is the gravitational acceleration (m/s$^2$); $f_{cor}$ is the Coriolis parameter; $\tau_{bx}$ and $\tau_{by}$ are shear stresses (Pa) on the bed surface and were calculated as follows:

$$\frac{\tau_{b(x,y)}}{\rho} = \frac{gn^2}{h^{1/3}}\sqrt{u^2 + v^2}(u, v) \tag{4}$$

$n$ is Manning's roughness (m$^{-1/3}$s); $\tau_{wx}, \tau_{wy}$ are surface wind shear stresses (Pa):

$$(\tau_{wx}, \tau_{wy}) = \rho_{air}c_{fa}\sqrt{U_w^2 + V_w^2}(U_w, V_w) \tag{5}$$

where $c_{fa}$ is the friction coefficient at the water surface and $U_w$ and $V_w$ are wind velocity (m/s); and, $\tau_{xx}, \tau_{xy}, \tau_{yx}$, and $\tau_{yy}$ are the depth-integrated Reynolds stresses (Pa) including both viscous and turbulent effects and approximated based on the Bousssinesq assumption:

$$\tau_{xx} = 2\rho(\nu + \nu_t)\frac{\partial u}{\partial x} \tag{6a}$$

$$\tau_{xy} = \tau_{yx} = \rho(\nu + \nu_t)\left(\frac{\partial u}{\partial y} + \frac{\partial v}{\partial x}\right) \tag{6b}$$

$$\tau_{yy} = 2\rho(\nu_t + \nu)\frac{\partial v}{\partial y} \tag{6c}$$

where a mixing-length model [21] was adopted to calculate the eddy viscosity $\nu_t$ (m$^2$/s).

### 2.2. Sediment-Transport Model

In this study, a single-phase sediment-transport model was developed, mainly consisting of the suspended-load transport equation, the bed-load transport equation, and the bed-change equation.

#### 2.2.1. Suspended-Load Transport

For suspended loads, the governing equation for the *k*th size class reads [19]

$$\frac{\partial (hC_k/\beta_{sk})}{\partial t} + \frac{\partial (huC_k)}{\partial x} + \frac{\partial (hvC_k)}{\partial y} = \frac{\partial}{\partial x}[h(\varepsilon_s \frac{\partial (C_k)}{\partial x} + D_{sxk})] + \frac{\partial}{\partial y}[h(\varepsilon_s \frac{\partial (C_k)}{\partial y} + D_{syk})] + \alpha \omega_{sk}(C_{*k} - C_k) \ (k = 1, 2, \ldots, n_s) \tag{7a}$$

where $C$ is the depth-averaged suspended-load concentration by volume (m$^3$/m$^3$), the concentration by mass is therefore $\rho_s C_k$ (kg/m$^3$) with $\rho_s$ the sediment density (kg/m$^3$); $C_{*k}$ is the suspended-load transport capacity; $\omega_{sk}$ is the settling velocity (m/s); $\alpha$ is the adaptation coefficient for the suspended load; $n_s$ is the number of size classes; the correction factor

$$\beta_{sk} = (\int_{z_b+\delta}^{\eta} u_s C_k dz)/(U_s \int_{z_b+\delta}^{\eta} C_k dz) = \overline{U_s}/U \tag{7b}$$

represents the lag between the flow and the suspended-load transport ($\leq 1$) with $\delta$ the thickness of the bed-load zone near the bed, $u_s$ the velocity of the suspended load (m/s), and $U_s$ the velocity magnitude (m/s). The suspended-load diffusivity coefficient is $\varepsilon_s = \nu_t/\sigma_c$, with the Schmidt number $0.5 \leq \sigma_c \leq 1$ and the dispersion terms $D_{sx} = -\frac{1}{h}\int_{z_b}^{\eta} (u_{3d} - u_{2d})(c_{3d} - C_{2d})dz$ and $D_{sy} = -\frac{1}{h}\int_{z_b}^{\eta} (v_{3d} - v_{2d})(c_{3d} - C_{2d})dz$ account for the nonuniform distributions of flow velocity and sediment concentration over flow depth, with small $u_{3d}$, $v_{3d}$, and $c_{3d}$ in 3D. In this study, the dispersion terms were ignored so that $D_{sx} = 0$ and $D_{sy} = 0$.

In Equation (7), there is one calibration parameter, the adaptation coefficient $\alpha$ for the suspended load, and three input variables, including the settling velocity $\omega_s$, the suspended-load transport capacity $C_*$, and the correction factor $\beta_s$, to be determined by semiempirical formulas. As mentioned previously, there are so many formulas available with their own advantages, limitations, and application ranges, this study does not focus on the comparisons of those formulas for "best picks"; we selected appropriate formulas that provided reasonable results during the validation process.

For the settling velocity, Zhang's formula [18] for naturally-worn sediment particles was used, which assumes a combination of drag force in the laminar region and the turbulent region.

$$\omega_s = \sqrt{(13.95\frac{\nu}{d})^2 + 1.09(\frac{\rho_s}{\rho} - 1)gd} - 13.95\frac{\nu}{d} \tag{8}$$

where $d$ is the particle diameter (m) and $\nu$ is the kinematical viscosity (m$^2$/s). Julien [37] and Cheng [38] also proposed similar formulas as Equation (8).

A variety of formulas have been developed for the suspended-load transport capacity. For example, Einstein [39] determined the suspended-load transport rate by integrating the local sediment concentration over the suspended-load zone; Zhang [18], based on the energy balance of sediment-laden flow, derived the relation between suspended-load transport capacity $C_*$ and parameter $U^3/(gR\omega_s)$; and Bagnold's formula [40] was based on

the stream power concept. The formula proposed by Wu et al. [41], based on the stream power concept, was selected:

$$q_{s*k} = 0.0000262[(\frac{\tau}{\tau_{ck}} - 1)\frac{U}{\omega_{sk}}]^{1.74} \cdot p_{bk}\sqrt{(\gamma_s/\gamma - 1)gd^3} \qquad (9)$$

where $q_{s*k}$ is the fractional suspended transport capacity (m$^2$/s); $\gamma_s$ and $\gamma$ are the specific weight (N) of sediment and water, respectively; $p_{bk}$ is the fraction of $k$th size class; and $\tau$ is the shear stress of entire cross section (Pa); $\tau_{ck} = 0.03(\gamma_s - \gamma)d_k(p_{hk}/p_{ek})^{0.6}$ with the hiding probability $p_{hk} = \sum_j p_{bj}d_j/(d_j + d_k)$ and the exposure probability $p_{ek} = \sum_j p_{bj}d_k/(d_j + d_k)$.

The correction factor $\beta_s$, the lag between the flow and sediment velocities, can be ignored for fine sediments, though for coarse sediments it is nonnegligible. According to [42], based on the logarithmic velocity distribution and the Rouse distribution for suspended load, the correction factor relation to the Rouse number $R$ and the Chezy's coefficient $C_h$ is defined as:

$$\beta_s = \begin{cases} 0.0289R^3 - 0.0448R^2 - 0.2977R + 0.994 \ (C_h = 40) \\ 0.0222R^3 - 0.0433R^2 - 0.1825R + 0.9924 \ (C_h = 60) \\ 0.0189R^3 - 0.044R^2 - 0.122R + 0.9924 \ (C_h = 80) \\ 0.0113R^3 - 0.0227R^2 - 0.1051R + 0.9918 \ (C_h = 100) \end{cases} \qquad (10)$$

where $R = \omega_s/(\kappa U_*)$ is the Rouse number with $\kappa$ the van Karman constant (=0.41) and $U_* = \sqrt{\tau_0/\rho}$ is the shear velocity (m/s). Interpolation may be performed with Equation (10) for the other $C_h$ values.

### 2.2.2. Bed-Load Transport

The governing equation for bed loads of $k$th size class [19] reads as follows:

$$\frac{\partial(q_{bk}/u_{bk})}{\partial t} + \frac{\partial(\alpha_{bx}q_{bk})}{\partial x} + \frac{\partial(\alpha_{by}q_{bk})}{\partial y} = \frac{1}{L_b}(q_{b*k} - q_{bk}) \qquad (11)$$

where $q_{bk}$ is the bed-load transport rate by volume per unit time and width (m$^2$/s); $q_{b*k}$ is the transport capacity (m$^2$/s); $L_b$ is the bed-load adaptation length (m); and the direction cosines of bed-load movement are $\alpha_{bx} = u/U$ and $\alpha_{by} = v/U$ with $U$ the velocity magnitude (m/s) when the effect of the bed slope is ignored.

To solve Equation (11), the bed-load velocity formula established by van Rijn [43] and the bed-load transport capacity formula developed by Wu et al. [41] were adopted. For the bed-load velocity, van Rijn [43] proposed the concept of the transport stage number $T$, which was defined as the excess bed-shear stress $T = (U'_*/U_{*cr})^2 - 1$, with $U'_* = Ug^{0.5}/[18\log(4h/d_{90})]$ the effective shear velocity and $U_{*cr}$ the critical shear velocity given by the Shields diagram. Then, the bed-load velocity is estimated by

$$u_b = 1.5T^{0.6}\sqrt{(\rho_s/\rho - 1)gd}. \qquad (12)$$

The bed-load transport capacity is related to the flow conditions and sediment properties. The formulas for bed-load transport capacity can be categorized into stream-power based (or velocity based) (i.e., Ref. [40]), shear-stress based (i.e., Refs. [44–46]), and probability-theory based (i.e., Ref. [38]). Wu et al. [41] proposed the use of the nondimensional excess grain shear stress $T_k = (n'/n)^{3/2}\tau_b/\tau_{ck} - 1$ for the estimation

$$q_{b*k} = 0.0053[(\frac{n'}{n})^{3/2}\frac{\tau_b}{\tau_{ck}} - 1]^{2.2} \cdot p_{bk}\sqrt{(\gamma_s/\gamma - 1)gd_k^3} \qquad (13)$$

where $q_{b*k}$ is the fractional bed-load transport capacity (m$^2$/s); $\tau_{ck}$ is the critical shear stress (Pa); $n' = d_{50}^{1/6}/20$, and $n$ is the Manning's roughness coefficient (m$^{-1/3}$s). Note that the hiding and exposure effect in nonuniform bed material is considered through $\tau_{ck}$.

In Equations (7) and (11), the adaptation coefficient, $\alpha$, for suspended load, and the adaptation length, $L_b$, for bed load, are critical to predicting nonequilibrium sediment transport. Physically, the adaptation length $L_b$ represents the distance required for the sediment transport to move from nonequilibrium to equilibrium conditions, while the adaptation coefficient $\alpha$ originally is related to the near-bed suspended-load concentration $C_b$, with $C_b = \alpha C$. To be consistent, an adaptation length for the suspended load (m) can also be defined as $L_s = Uh/(\alpha \omega_s)$. In practice, $\alpha$ and $L_b$ are case-dependent calibration parameters. The adaptation length for the bed load, $L_b$, is closely related to the bed-form characteristics and can be evaluated using the characteristic lengths, such as channel width, sand-dune length, alternate-bar length, water depth, and mesh-cell length. The adaptation coefficient $\alpha$ is in the range of [0, 1]. Based on the work of Galappatti and Vreugdenhil [47], Armanini and de Silvio [48] proposed the following function to evaluate $\alpha$:

$$\alpha = 1/\left\{\frac{a}{h} + \left(1 - \frac{a}{h}\right)\exp\left[-1.5\left(\frac{a}{h}\right)^{-1/6}\frac{\omega_s}{U_*}\right]\right\} \tag{14}$$

where $a$ is related to the zero-velocity distance in the logarithmic velocity distribution, defined as $a = 33h/\exp(1 + \kappa C_h/\sqrt{g})$.

The suspended-load transport Equation (7) and the bed-load transport Equation (11) can be combined into a total-load transport equation [20,26]. The single total-load equation is computationally more convenient since only one equation is solved, though it requires an additional fraction parameter to identify the suspended load and the bed load in the total load. In this study, however, the suspended-load equation and the bed-load equation will be solved separately, which is conceptually simple and clear.

### 2.2.3. Bed Changes and Sorting

The source terms on the right-hand side (RHS) of Equations (7a) and (11) represent the net exchange of sediment between the bed and the flowing water, including the suspended load in the water column and the bed load in the near-bed zone. Therefore, the bed changes can be calculated by

$$(1 - p'_m)\frac{\partial z_{bk}}{\partial t} = \alpha \omega_{sk}(C_k - C_{*k}) + \frac{1}{L_b}(q_{bk} - q_{b*k}) \tag{15}$$

where $p'_m$ is the sediment porosity, and $z_{bk}$ represents the bed elevation (m) contributed by $k$th size class.

In the vertical direction, the bed can be divided into at least three layers from top to bottom: the mixing layer (also called the active layer), the immediate subsurface layer, and the bottom subsurface layer. The bottom subsurface layer can be further divided into more sublayers. The exchanges of sediment between the water column and the bed leads to bed sorting with dynamic variations of sediment composition in each bed layer.

As is widely accepted, sediment exchange occurs at the mixing layer. The composition of the mixing layer can be found using a mass-balance approach (e.g., [19,25,26,49]). This study used the equation proposed by Wu [19] for the mixing layer.

$$\frac{\partial(A_m p_{bk})}{\partial t} = \frac{\partial z_{bk}}{\partial t} + p^*_{bk}\left(\frac{\partial A_m}{\partial t} - \frac{\partial z_b}{\partial t}\right) \tag{16}$$

where $A_m$ is the thickness of the mixing layer and $p_{bk}$ is the fraction of the $k$th size class in the mixing layer. When $\frac{\partial A_m}{\partial t} - \frac{\partial z_b}{\partial t} \geq 0$, bed changes remain in the current-mixing layer, so $p^*_{bk} = p_{bk}$; when $\frac{\partial A_m}{\partial t} - \frac{\partial z_b}{\partial t} < 0$, bed changes penetrate to the immediate subsurface layer, so $p^*_{bk} = p_{ik}$ (the fraction of $k$th size class in the immediate subsurface layer).

The thickness of the bed layers (the mixing layer, the immediate subsurface layer, and the bottom subsurface layer) are user-input variables for the simulation. For the mixing layer, a minimum thickness, $A_{\min}$, was set to maintain the active sediment exchanges in the mixing layer. This minimum thickness of the mixing layer, $A_{\min}$, can be either user-specified or determined by $d_{50}$ and the sand dune height $\Delta_s$ [20].

$$A_{\min} = \max(2d_{50}, 0.5\Delta_s) \tag{17}$$

The mixing-layer thickness changes little during the computation, though the division of each layer may move upward with net deposition or downward with net erosion. In such cases, the composition changes of the mixing layer may induce corresponding changes in the other subsurface layers. For example, for the immediate subsurface layer, one can obtain [20]

$$\frac{\partial(A_i p_{ik})}{\partial t} = -p_{bk}^*\left(\frac{\partial A_m}{\partial t} - \frac{\partial z_b}{\partial t}\right) \tag{18}$$

where $A_i$ is the thickness of the immediate subsurface layer.

For the nonerodible bed with only net depositions allowed, both erosions and the bed-layer thickness need to be limited.

## 3. Numerical Method

The governing equations of the flow (Equations (1)–(3) and sediment transport (Equations (7) and (11)) were discretized using the finite volume method (FVM) on a collocated unstructured hybrid-mesh system with mixed-triangle and quadrilateral cells. Compared to the structured-mesh system, the unstructured mesh is more suitable for geometrically complex domains with higher adaptivity.

### 3.1. FVM Discretization

Following the work of Zhang et al. [36], the discretization method is illustrated by the integral form of the suspended-load transport equation in the typical cells shown in Figure 1, where one triangle cell and one quadrilateral cell are neighbored with each other. In each cell, the centroid nodes (denoted by the superscript "c") and the vertex nodes (denoted by the superscript "v") are counterclockwise numbered. A similar discretization procedure can be carried out for the bed-load transport equation.

For the time integral, the first-order Euler forward scheme was used:

$$\int_\Omega \frac{\partial(hC_k/\beta_{sk})}{\partial t}d\Omega \approx \frac{(hC_k/\beta_{sk})^{n+1} - (hC_k/\beta_{sk})^n}{\Delta t} \cdot \Omega \tag{19}$$

where the superscript "n" denotes the time level.

For the convective fluxes,

$$\int_\Omega \left(\frac{\partial(uhC_k)}{\partial x} + \frac{\partial(vhC_k)}{\partial y}\right)d\Omega \approx \sum F_{0-m}(hC_k)_{0-m} \tag{20}$$

where $F_{0-m} = u_{0-m}\Delta y_{i-j} - v_{0-m}\Delta x_{i-j}$ is the flux; the subscripts "i" and "j" denote the two vertices of the edge "$0 - m$" ordered counterclockwise. Note that for coordinate calculations, the subscript "$i - j$" denotes the subtraction of coordinate "i" from "j"; that is, $\Delta x_{i-j} = x_j - x_i$ and $\Delta y_{i-j} = y_j - y_i$.

The second-order upwind scheme was used for the edge center "$0 - m$",

$$\phi_{0-m} \approx \begin{cases} \phi_0 + (\phi_x)_0\Delta x_{0-F} + (\phi_y)_0\Delta y_{0-F} & (F_{0-m} > 0) \\ \phi_m + (\phi_x)_m\Delta x_{m-F} + (\phi_y)_m\Delta y_{m-F} & (F_{0-m} < 0) \end{cases} \tag{21}$$

where $\phi_{0-m} = (hC_k)_{0-m}$ and, $\phi_x$ and $\phi_y$ represent the first-order derivatives with respect to $x$ and $y$.

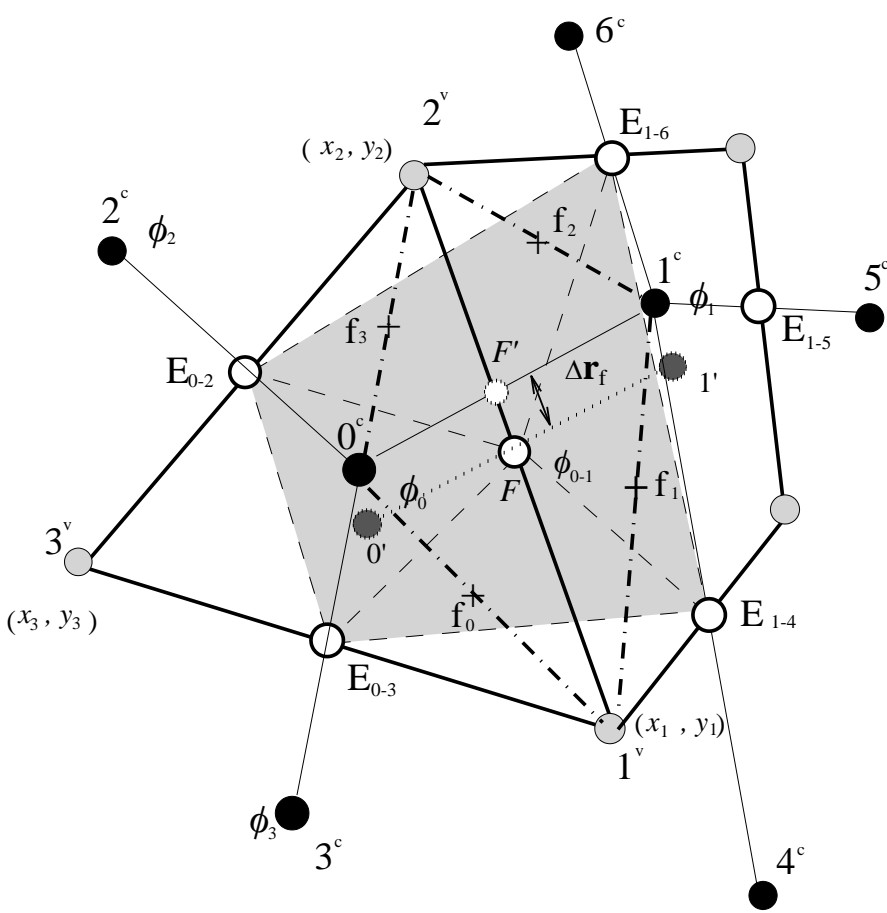

**Figure 1.** Typical triangle and quadrilateral cells: the black circle denotes the cell centers; the gray circle denotes the vertices; the big white circles denote the edge centers; the small white circle denotes the interception point of the edge and the line of two neighboring cell centers; the blank quadrilateral ($0^c$-$1^v$-$1^c$-$2^v$) with the dash-dot lines is formed by two vertices of the edge and two neighboring cell centers; and the quadrilateral filled with gray color is formed by four neighboring edge centers ($E_{0-1}$, $E_{0-3}$, $E_{1-4}$, and $E_{1-6}$) with $E_{0-1}$ as centroid.

In Equation (21), the first term on the RHS will be treated implicitly, while the second term with gradients is treated as the source term.

$$S^\phi = \sum_m \left\{ F_{0-m}^+ \cdot [(\phi_x)_0 \Delta x_{0-F} + (\phi_y)_0 \Delta y_{0-F}] + F_{0-m}^- \cdot [(\phi_x)_m \Delta x_{m-F} + (\phi_y)_m \Delta y_{m-F})] \right\} \tag{22a}$$

$$F_{0-m}^\pm = \frac{1}{2}(F_{0-m} \pm |F_{0-m}|) \tag{22b}$$

For the diffusion fluxes, the Green Theorem was applied so that one can obtain:

$$\int_{\Omega_{0-m}} \left[ \frac{\partial}{\partial x}\left(h\varepsilon_s \frac{\partial C_k}{\partial x}\right) + \frac{\partial}{\partial y}\left(h\varepsilon_s \frac{\partial C_k}{\partial y}\right) \right] d\Omega \approx \sum_m \left[ \left(h\varepsilon_s \frac{\partial C_k}{\partial x}\right)_{0-m} \cdot \Delta y_{i-j} - \left(h\varepsilon_s \frac{\partial C_k}{\partial y}\right)_{0-m} \cdot \Delta x_{i-j} \right] \tag{23}$$

where $\Omega_{0-m}$ is the area of the quadrilateral at the edge "$0-m$" (refer to the quadrilateral $0^c$-$1^v$-$1^c$-$2^v$ in Figure 1).

For the source terms, one can obtain:

$$\int_\Omega [\alpha\omega_{sk}(C_{*k} - C_k) - S^\phi] d\Omega \approx [-\alpha\omega_{sk}C_k + \alpha\omega_{sk}C_{*k} - S^\phi] \cdot \Omega \tag{24}$$

where the first term with $C_k$ at RHS will be treated implicitly.

### 3.2. Edge-Gradient Evaluation

On the RHS of Equation (23), the edge gradient was evaluated by the Green Theorem, and the values at the vertices are required, which are interpolated from the cell centers.

$$\left(\frac{\partial C}{\partial x}\right)_{0-m} = \frac{1}{\Omega_{0-m}} \oint C dy \approx \frac{1}{2\Omega_{0-m}} [(C_m^c - C_0^c)\Delta y_{i-j} - (C_j^v - C_i^v)\Delta y_{0-m}] \tag{25a}$$

$$\left(\frac{\partial C}{\partial y}\right)_{0-m} = \frac{-1}{\Omega_{0-m}} \oint C dx \approx \frac{-1}{2\Omega_{0-m}} [(C_m^c - C_0^c)\Delta x_{i-j} - (C_j^v - C_i^v)\Delta x_{0-m}] \tag{25b}$$

Note that the subscript "*k*" is omitted here for simplicity.

If the Taylor series expansion is used, no interpolation is needed, and one can obtain:

$$\left(\frac{\partial C}{\partial x}\right)_{0-m} = \frac{1}{\Omega_{0-m}} \oint C dy \approx \frac{1}{\Omega_{0-m}} [(C_m^c - C_0^c + D_{0-m})\Delta y_{i-j} + (G_x)_{0-m}] \tag{26a}$$

$$\left(\frac{\partial C}{\partial y}\right)_{0-m} = \frac{-1}{\Omega_{0-m}} \oint C dx \approx \frac{-1}{\Omega_{0-m}} [(C_m^c - C_0^c + D_{0-m})\Delta x_{i-j} + (G_y)_{0-m}] \tag{26b}$$

$$D_{0-m} = [(C_x)_m - (C_x)_0] \cdot \Delta x_{F-F'} + [(C_y)_m - (C_y)_0] \cdot \Delta y_{F-F'} \tag{26c}$$

$$(G_x)_{0-m} = \frac{1}{2}[(C_x)_0 \Delta x'_{0-i} + (C_y)_0 \Delta y'_{0-i}] \cdot \Delta y'_{0-i} + \frac{1}{2}[(C_x)_0 \Delta x'_{0-j} + (C_y)_0 \Delta y'_{0-j}] \cdot \Delta y'_{j-0} + \\ \frac{1}{2}[(C_x)_m \Delta x'_{m-i} + (C_y)_m \Delta y'_{m-i}] \cdot \Delta y'_{i-m} + \frac{1}{2}[(C_x)_m \Delta x'_{m-j} + (C_y)_m \Delta y'_{m-j}] \cdot \Delta y'_{j-m} \tag{26d}$$

$$(G_y)_{0-m} = \frac{1}{2}[(C_x)_0 \Delta x'_{0-i} + (C_y)_0 \Delta y'_{0-i}] \cdot \Delta x'_{0-i} + \frac{1}{2}[(C_x)_0 \Delta x'_{0-j} + (C_y)_0 \Delta y'_{0-j}] \cdot \Delta x'_{j-0} + \\ \frac{1}{2}[(C_x)_m \Delta x'_{m-i} + (C_y)_m \Delta y'_{m-i}] \cdot \Delta x'_{i-m} + \frac{1}{2}[(C_x)_m \Delta x'_{m-j} + (C_y)_m \Delta y'_{m-j}] \cdot \Delta x'_{j-m} \tag{26e}$$

where $\Omega_{0-m}$ denotes the areas of the quadrilateral at the edge $0 - m$; $C_x$ and $C_y$ denote the first-order derivatives; $D_{0-m}$ is the correction term for mesh irregularity (will automatically vanish in well-connected mesh), and all the coordinates with the superscript "'" count in the distance vector $\nabla \vec{r}_f$ as well. For example, $\Delta x'_{0-i} = x_i^v - (x_0^c + \Delta x_{F-F'})$ and $\Delta y'_{m-i} = y_i^v - (y_m^c + \Delta y_{F-F'})$, and so on.

This evaluation method for the edge gradient was also applied to the velocity gradient at edges in the momentum equations and the pressure gradient at edges when coupling the continuity equation and the momentum equations. Compared to the conventional method requiring vertex values, this method accounts for mesh irregularities and avoids interpolations for the vertex values.

### 3.3. Multipoint Momentum Interpolation Correction

After discretization, the momentum Equations (2) and (3) can be linearized as follows:

$$\frac{A^{U*}}{r_u}U^* + \sum_1^b A_b^U U_b^* = U^n - g\Delta t(\nabla \eta) + \frac{(1-r_u)}{r_u}A^{U*}U^m + S^U \tag{27}$$

where $U^* = (u^*, v^*)$ is the provisional velocity (m/s); $U^m$ is the velocity of the previous iteration (m/s) with the superscript "*m*" denoting the iteration step; $r_u$ is the under-relaxation parameter for velocity; $\nabla \eta = (\partial \eta / \partial x, \partial \eta / \partial y)$ is the pressure gradient; $A^{*U} = (A^{*u}, A^{*v})$ is the nondimensional matrix coefficient; $A_b^U = (A_b^u, A_b^v)$ is the matrix coefficient for neighboring cells; and $S^U$ is the source term.

For the collocated mesh system, the momentum interpolation (MI) method proposed by Rhie and Chow [50] is to interpolate the discretized momentum equation at the edge between two neighboring cells to evaluate the edge velocity and flux.

$$u_{0-m}^* \approx IS_{0-m} - \frac{g\Delta t}{A_{0-m}^U}(\nabla \eta)_{0-m} \tag{28}$$

where $IS = U^* + \frac{g\Delta t}{A^U}(\nabla\eta)$ denotes the interpolation subject (m/s) at cell centers; $IS_{0-m}$ denotes the cross-edge interpolation subject with $IS_{0-m} = s_{0-m}IS_0 + (1 - s_{0-m})IS_m$ with $s_{0-m}(\in [0,1])$ the linear interpolation coefficient; and the matrix coefficient at the edge is defined as $A^U_{0-m} = s_{0-m} \cdot A^U_0 + (1 - s_{0-m}) \cdot A^U_m$.

Zhang and Jia [51] extended the cross-edge MI to the surrounding edge centers by defining the multipoint interpolation subject as the averaged surrounding edge interpolation subject $IS^{MP}_{0-m} = \sum_1^4 w_n IS_{i-j}$ with $w$ the weighting coefficient. Specifically, at edge "0–1" in Figure 1, the multipoint interpolation subject is $IS^{MP}_{0-1} = w_1 IS_{0-1} + w_2 IS_{0-2} + w_3 IS_{1-4} + w_4 IS_{1-6}$ with the superscript "MP" representing "multi-point". Therefore, the momentum interpolation with multipoint interpolation correction reads:

$$U^*_{0-m} \approx IS_{0-m} - \frac{g\Delta t}{A^U_{0-m}}(\nabla\eta)_{0-m} + (IS^{MP}_{0-m} - IS_{o-m}) \cdot r_I \tag{29}$$

where $r_I$ is a relaxation factor in the range of [0, 1] to control the correction of the last term on the RHS of Equation (29). With the multipoint-momentum interpolation corrections, nonphysical oscillations can be removed, especially those possible oscillations induced by the wetting-and-drying process, which is common due to the morphological changes in the unsteady sediment transport.

### 3.4. Solution Procedure

All discretized equations are solved by the BiCGSTAB(*l*) (biconjugate-gradient stabilized method) solver [52]. At each time step, the flow and the sediment transport are coupled in the following way: (1) the flow field is calculated first based on the current bed conditions (bed elevation and bed material composition); (2) then, the sediment transport is simulated using the calculated flow field; and (3) finally, the bed changes and the bed sorting are calculated and bed elevations and sediment compositions are updated.

## 4. Examples and Application

In the following sections, the hybrid-sediment transport model will be demonstrated and validated by selected examples, including the degradation case [9] for the erosion process, the aggradation case [53] for the deposition process, and the long-term unsteady sediment transport in the East Fork River [54]. Finally, the model was applied to JiJi Reservoir in Taiwan [1]. All computational meshes were generated by CCHE-MESH [55], a quality mesh generator for both structured and unstructured meshes.

### 4.1. Bed Degradation

The first example is based on the benchmark experiments conducted by Ashida and Michiue [9] to demonstrate the model's capability of handling bed degradation and armoring processes under clear-water conditions. The experimental flume was 20 m long and 0.8 m wide with a slope of 1%. On the flume bed, there were 12 size classes of sediment particles (Table 1) ranging from sand to gravel with the median size of 1.5 mm and the initial bed material thickness of 0.113 m. A steady flow of 0.0314 m³/s was imposed at the inlet, while the water level of 0.06 m was maintained at the outlet.

In this simulation, the bed-load transport dominated, and the bed-load transport model was run on a $5 \times 100$ rectangle mesh. Bed roughness was calibrated as 0.023 $m^{-1/3}s$, and the time step was 1 s. For this bed degradation case, Wu [20] investigated the effects of the mixing layer thickness (varying between $d_{50}$ and $2d_{50}$) and the bed-load adaptation length (time dependent, water depth dependent, or constant) on the simulations using the one-dimensional model. In this study, the mixing-layer thickness was set as $2d_{50}$ and the bed-load adaptation length was set as the averaged sand dune length ($\approx 7.3$ times the water depth).

**Table 1.** Size classes for degradation experiment (Ashida and Michiue [3]).

| Size (mm) | 0.25 | 0.35 | 0.5 | 0.7 | 0.9 | 1.25 | 1.75 | 2.5 | 3.5 | 5 | 7 | 9 |
|---|---|---|---|---|---|---|---|---|---|---|---|---|
| Lower Bound | 0.2 | 0.3 | 0.4 | 0.6 | 0.8 | 1 | 1.5 | 2 | 3 | 4 | 6 | 8 |
| Upper Bound | 0.3 | 0.4 | 0.6 | 0.8 | 1 | 1.5 | 2 | 3 | 4 | 6 | 8 | 10 |
| Fraction | 0.075 | 0.125 | 0.165 | 0.035 | 0.035 | 0.065 | 0.04 | 0.09 | 0.1 | 0.195 | 0.05 | 0.025 |

With the clear water condition, bed erosion occurred due to bed-load sediment transport. During the scouring process, a layer of coarse bed materials (called an armoring layer) may be formed, which could protect the bed from scouring, resulting in slowing or even stopping erosion (called armoring effects). In this bed degradation case, the armoring effects were significant. Figure 2 compares the measured erosion depth development with time at $x$ = 7 m, 10 m, and 13 m to the simulation results. As can be seen, the erosion developed rapidly in the first 100 min and then became much milder afterward due to the armoring effects. The simulated erosion development profiles at three locations agreed well with the measured ones.

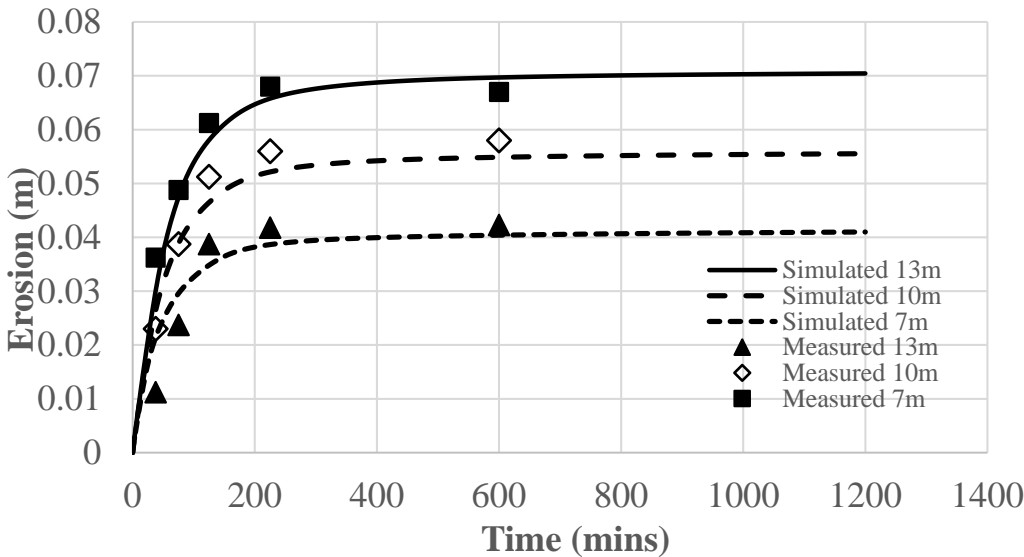

**Figure 2.** Erosion development profiles with time.

The armoring effect is closely related to the bed sorting calculation. Figure 3 compares the simulated bed composition with the measured one at $x$ = 10 m. The model succeeded in simulating the armoring effects. The initial $d_{50}$ was 1.5 mm and the measured final $d_{50}$ was 5.4 mm, while the simulated final $d_{50}$ was 5.6 mm. However, the simulated armoring layer was formed earlier, resulting in a slightly coarser bed-armoring layer. In general, the bed-sorting calculation was satisfactory for this case.

*4.2. Bed Aggradation*

The benchmark experiments conducted by Seal et al. [53] were selected to demonstrate the bed aggradation process. The experiment was designed to investigate longitudinal deposition formed by feeding poorly sorted sediment from upstream. The flume was 45 m long and 0.3 m wide with a slope of 2%. The input-sediment particle size varied from 0.125 mm (fine sand) to 64 mm (coarse gravel) with a median size of 6 mm. Figure 4 shows a sketch of this flume experiment.

In the experiment, there were three runs, and run two was selected in this study. For run two, at the inlet, a constant flow discharge of 0.049 m$^3$/s and a constant sediment discharge of 0.0942 kg/s were imposed, while at the outlet, the water level was 0.45 m. Table 2 lists the input sediment mixture consisting of 10 size classes. The sediment transport model ran for 32.4 h on a 5 × 200 rectangle mesh with the bed roughness of 0.025 m$^{-1/3}$s

and a time step of 1 s. The adaptation coefficient of the suspended load, $\alpha$, was calibrated as 0.3, while the adaptation length of the bed load, $L_b$, was 0.3 m (the channel width).

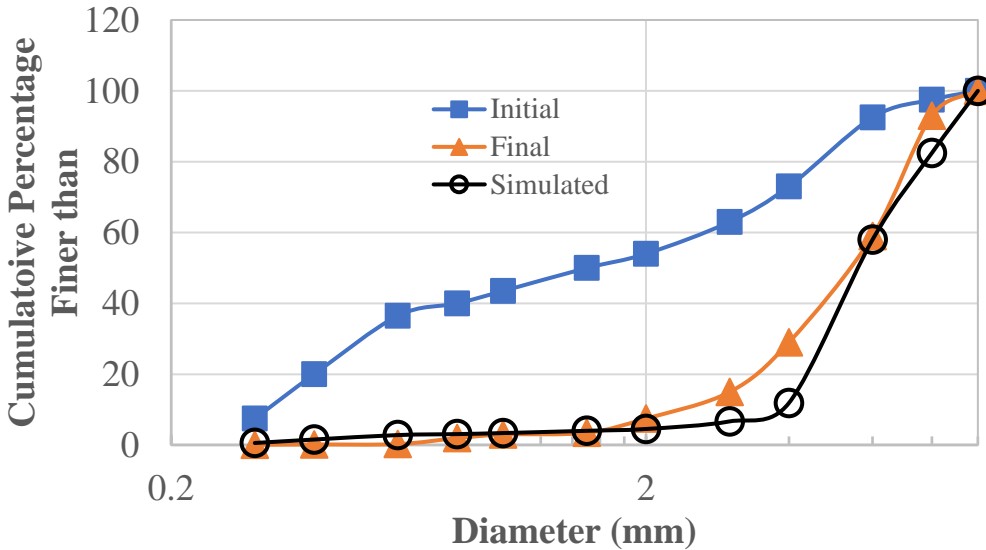

**Figure 3.** Bed compositions at $x$ = 10 m.

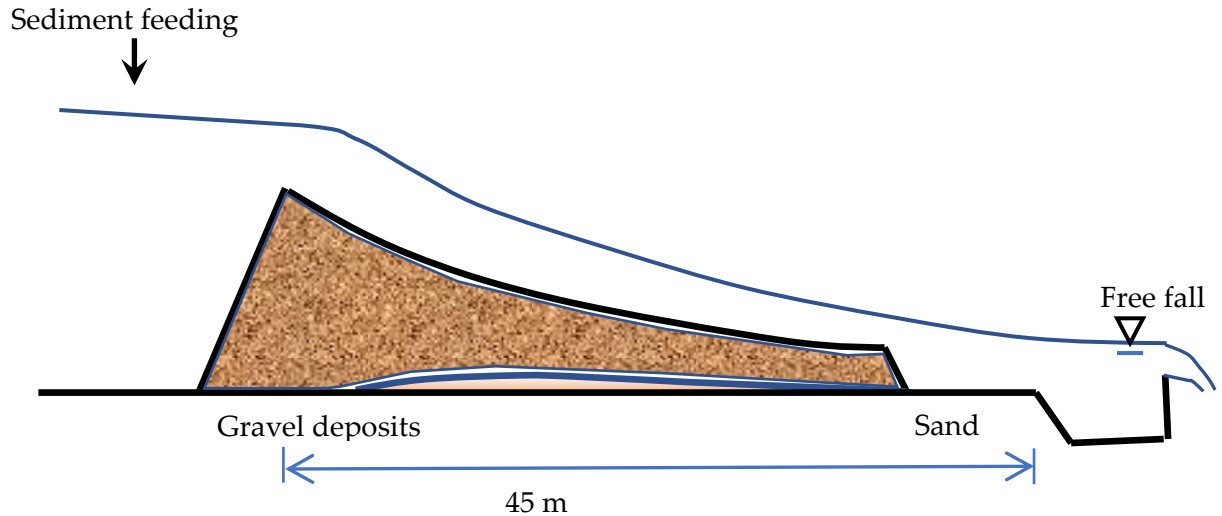

**Figure 4.** Sideview of the layout of the bed-aggradation experiment by Seal et al. [51].

**Table 2.** Input sediment mixture for the bed aggradation experiment (Seal et al. [32]).

| Size (mm) | 0.125 | 0.25 | 0.5 | 1 | 2 | 4 | 8 | 16 | 32 | 64 |
|---|---|---|---|---|---|---|---|---|---|---|
| Fraction | 0.022 | 0.104 | 0.102 | 0.07 | 0.04 | 0.172 | 0.12 | 0.168 | 0.124 | 0.078 |

Due to the constant feeding of coarse sediments, sediment deposition developed in this flume. Figure 5 compares the simulated bed profiles at different times to the measured ones. The shape of the bed profiles, including the ending slopes, was well captured by the model, though a small phase shift at 22 h was observed, resulting in an over-estimated deposition. However, the simulation results generally agreed well with the measurements.

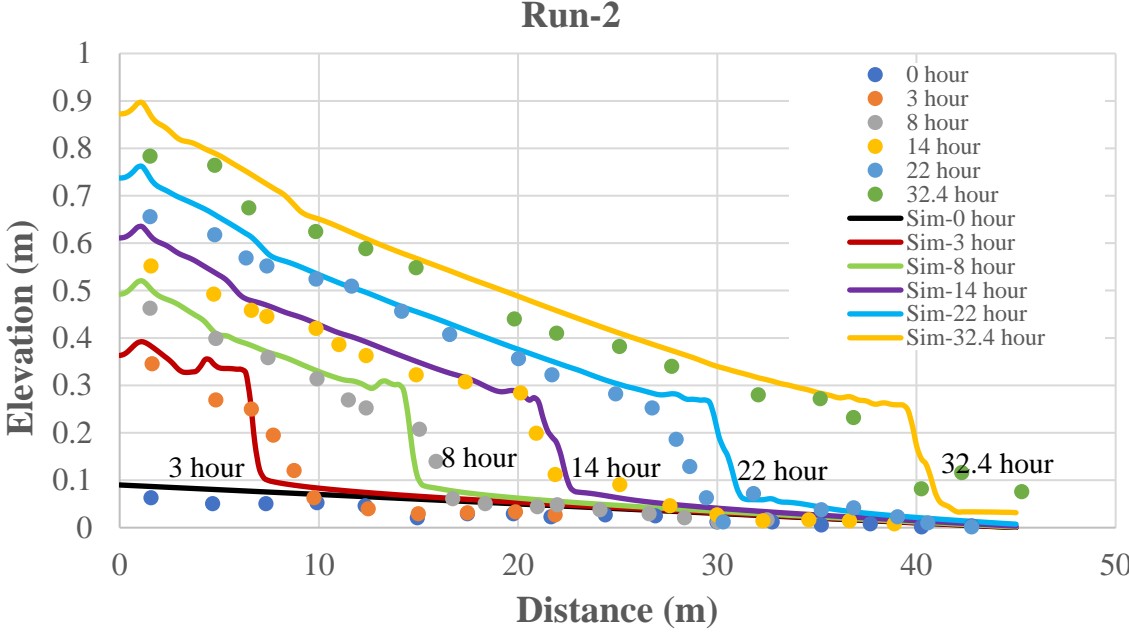

**Figure 5.** Longitudinal bed profiles at different times.

### 4.3. East Fork River

The East Fork River is a typical meandering river located in Wyoming State that was used for the study of bed-load transport in the 1970s. The selected reach was about 3.3 km long with the channel width varying from 16 m to 45 m and ending at a bed-load trap across the river (Figure 6). The flow in this river was influenced by spring runoff due to snowmelt, with rising flow rates in the morning, peak rates at midday, and declining flows in the afternoon. This field case was selected to demonstrate the capability of the model for long-term sediment transport under unsteady turbulent-flow conditions.

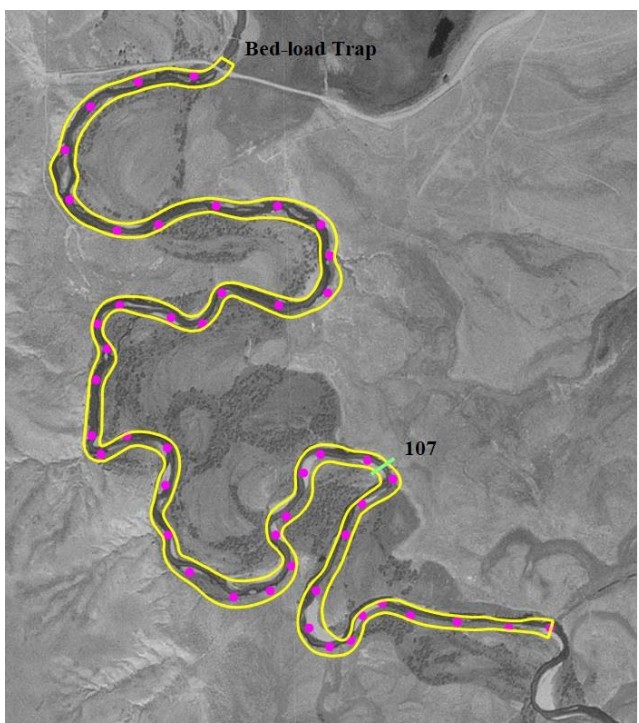

**Figure 6.** Selected reach of East Fork River. Solid circles represent the approximate locations of bed samples.

The simulation period was 2–19 June 1979. Figure 7 shows the hydrograph and the stage graph at the inlet and outlet. This model generated an unstructured quadrilateral mesh of this reach with minimum and maximum edge lengths equal to 0.46 m and 13.99 m (Figure 8). With the variation of flow, the wetting-and-drying process developed with time in this study reach, which may have induced nonphysical numerical oscillations in the model. The multipoint momentum interpolation correction technique [51] was used to remove the oscillations and maintain numerical stability and accuracy.

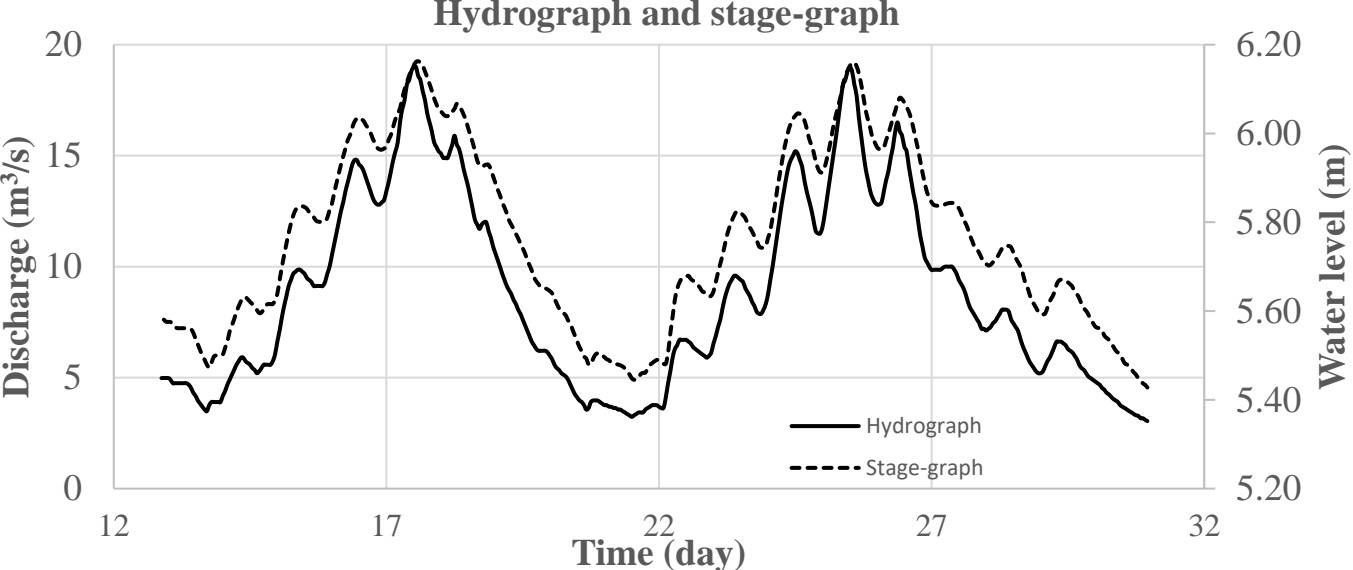

**Figure 7.** Hydrograph at the inlet and stage graph at the outlet.

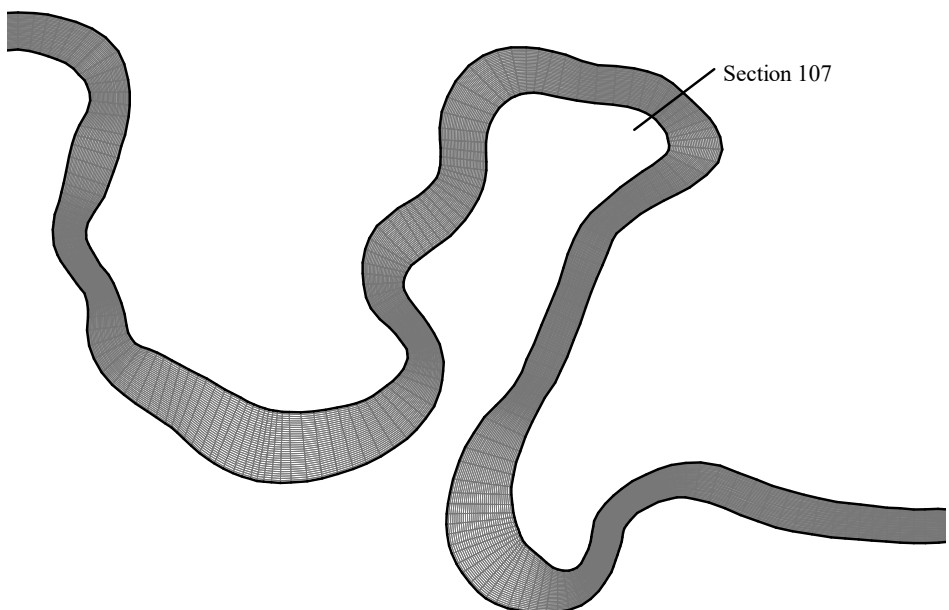

**Figure 8.** Local view of unstructured quadrilateral mesh with 16,533 cells and 16,000 nodes.

The measured water surface elevation at section 107 was used to calibrate the Manning coefficient, and Figure 9 compares the simulated and measured water surface profiles at section 107 using the calibrated Manning coefficient of 0.03 $\text{m}^{-1/3}\text{s}$ for the whole reach and the time step of 3 s. As can be seen, the simulated profile matched well with the measured one.

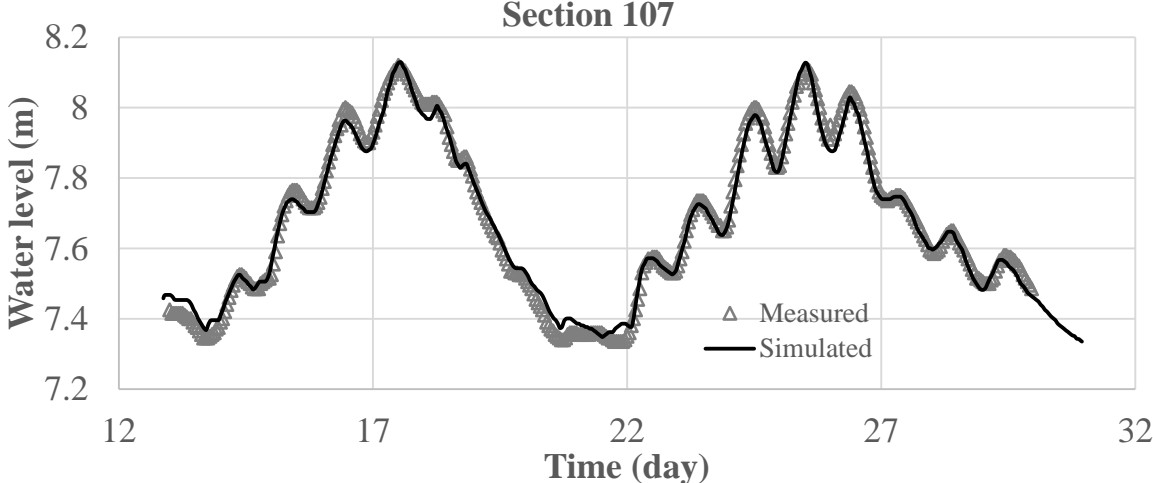

**Figure 9.** Water surface profiles with time at section 107.

Table 3 lists the initial nine size classes of sediment mixtures ranging from 0.088 mm (fine sand) to 32 mm (coarse gravel). Bed-sample information was obtained from the USGS technical report [52] for the selected reach. Although the bed load was dominant, both the suspended-load and bed-load transport were simulated. In this simulation, the adaptation coefficient of the suspended load, $\alpha$, was calibrated as 0.001, while the adaptation length of the bedload, $L_b$, was 60 m, about two times the average channel width.

**Table 3.** Sediment size classes for the East Fork River (Meade et al. [52]).

| Size (mm) | 0.088 | 0.177 | 0.354 | 0.707 | 1.41 | 2.83 | 5.66 | 11.3 | 32 |
|---|---|---|---|---|---|---|---|---|---|
| Fraction | 0.044 | 0.00038 | 0.02 | 0.478 | 0.233 | 0.145 | 0.093 | 0.02 | 0.01 |

At the outlet, the measured sediment flux at the bed trap was available to compare with the simulated one (Figure 10). According to the comparisons, the sediment-transport model captured the general trend of the variation of the sediment discharge with time, except for the overestimation of the first peak and the underestimation of the second peak. The simulation results were also identical to the ones from Wu [19].

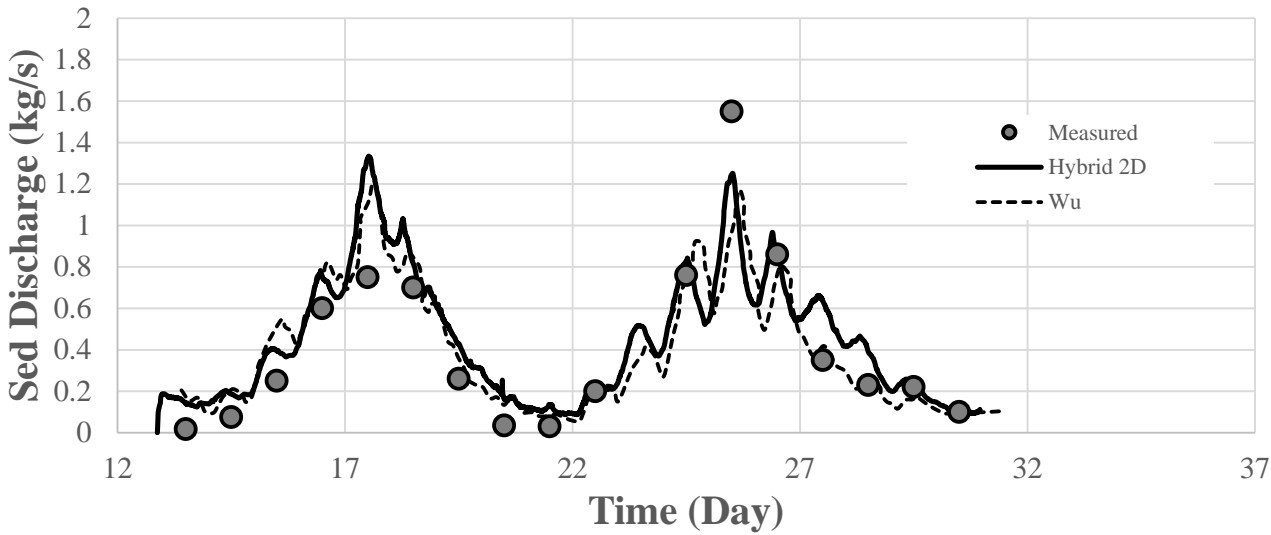

**Figure 10.** Sediment discharge at the outlet.

### 4.4. JiJi Reservoir

The sediment-transport model was used to simulate the deposition process in the JiJi Reservoir (Figure 11) in Taiwan, which was previously simulated by the CCHE2D sediment-transport model using structured meshes [1]. The JiJi Weir (Figure 11) was built across a mountain river, Chuoshui Creek, in Taiwan. The flow pattern of Chuoshui Creek was strongly affected by the precipitation pattern, with a very small discharge in dry seasons and extremely large flows in typhoon seasons. Due to the steep slopes and exposed soils, the flood water could transport high sediment loads and deposit them in the reservoir, resulting in rapid reductions in reservoir storage capacity. Once the reservoir was full, the sediment would be flushed through the spillway of the weir to the downstream channel.

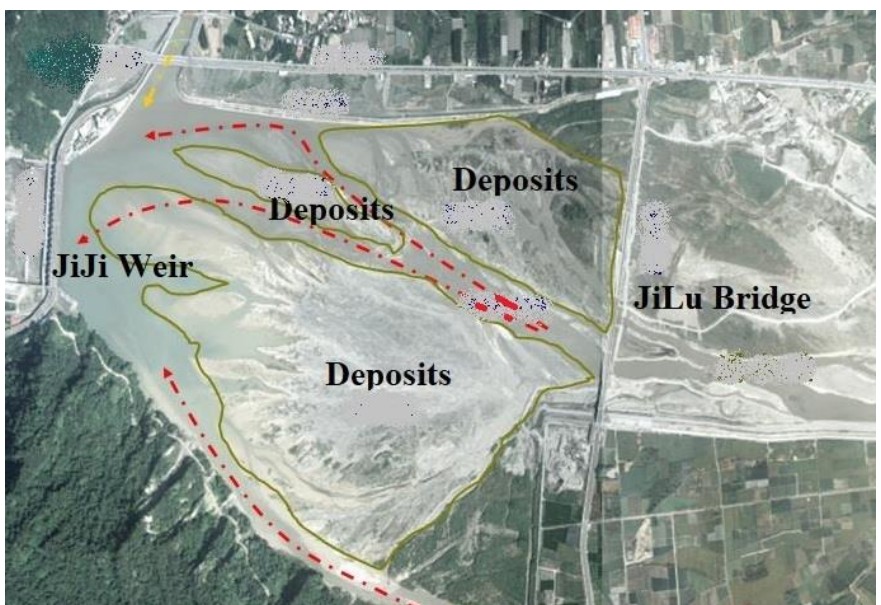

**Figure 11.** JiJi Reservoir.

Figure 11 shows the study domain from JiLu Bridge to JiJi Weir, which was about 2 km long. As can be seen, the reservoir suffered from serious sedimentation problems. In 2003, the total amount of sediment yield was about 14.77 million m$^3$, though only about 13% was transported downstream. According to the bed samples measured in July of 2004 (Figure 12), coarse sediments were deposited mainly upstream of the reservoir, close to the JiLu bridge, while fine sediments were deposited within about 1000 m of the weir.

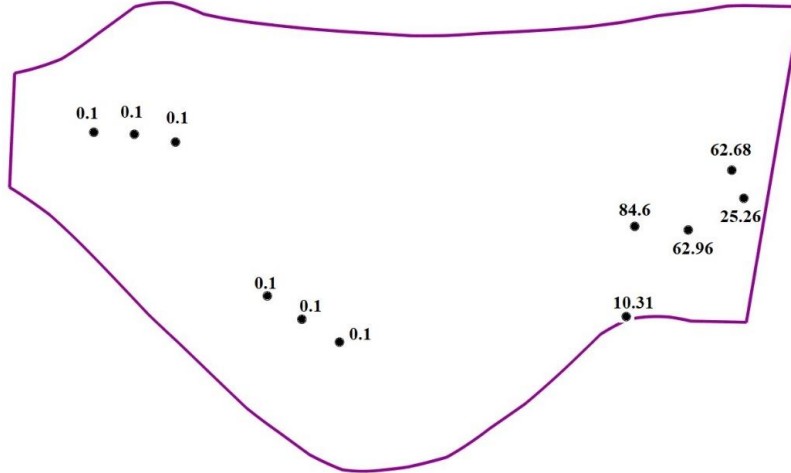

**Figure 12.** Bed sample locations with $d_{50}$ in mm.

Table 4 lists six sediment-size classes used in the simulation, varying from medium silts to small boulders. The bed samples of 2004 (Figure 12) were used as the initial bed composition within the reservoir. Figure 13 shows the initial bed elevation and the quadrilateral mesh generated for this reservoir based on the measured topography data in 2004.

**Table 4.** Size classes in the JiJi Reservoir.

| Size (mm) | 0.0272 | 0.297 | 2.38 | 9.52 | 152 | 457 |
| --- | --- | --- | --- | --- | --- | --- |
| Lower Bound | 0.001 | 0.074 | 0.59 | 4.76 | 19.1 | 305 |
| Upper Bound | 0.074 | 0.59 | 4.76 | 19.1 | 305 | 610 |

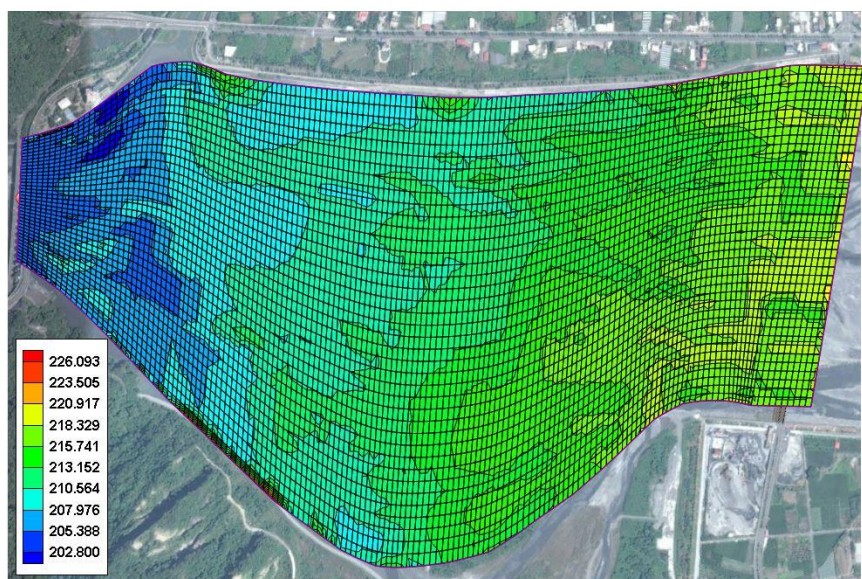

**Figure 13.** Initial bed elevations and computational mesh for the JiJi Reservoir with 6440 nodes and 6240 cells.

The simulation period was from July 2004 to October 2007. Due to the flow pattern of Chuoshui Creek, small discharges were assumed to have little effect on the sediment transport in the reservoir, which were therefore ignored in the simulation. The actual hydrograph and sedigraph consisted of all ten typhoon events during the study period with peak flow discharges varying from 1020 m³/s to 8343 m³/s, as shown in Figure 14a. Figure 14b shows the corresponding stage hydrograph at JiJi Weir. Such treatment shortened the simulation time from 3 years to 33 days.

Both suspended-load and bed-load transport models were applied to this reservoir with the adaptation coefficient of the suspended load, $\alpha$, set as 0.01, and the adaptation length of bed load $L_b$ = 4000 m.

Figure 15a shows the measured bed change pattern from 2004 to 2007. At the downstream end, sediment deposits approached the weir and formed a large hump across the reservoir, though a large scour hole also developed about 100 m from the weir. In the middle, sediment transport followed a general pattern of erosion on the left and deposition on the right, while erosion dominated upstream close to the JiLu Bridge. Note that the erosion on the left of the deposition delta (Figure 11) was partially caused by dredging activities with unknown amounts ([1,2]). Comparing to the measured values, the simulated bed changes (Figure 15b) captured the humps near the weir and the general deposition pattern on the right but missed the hole near the weir and predicted slight deposition on the left, as expected.

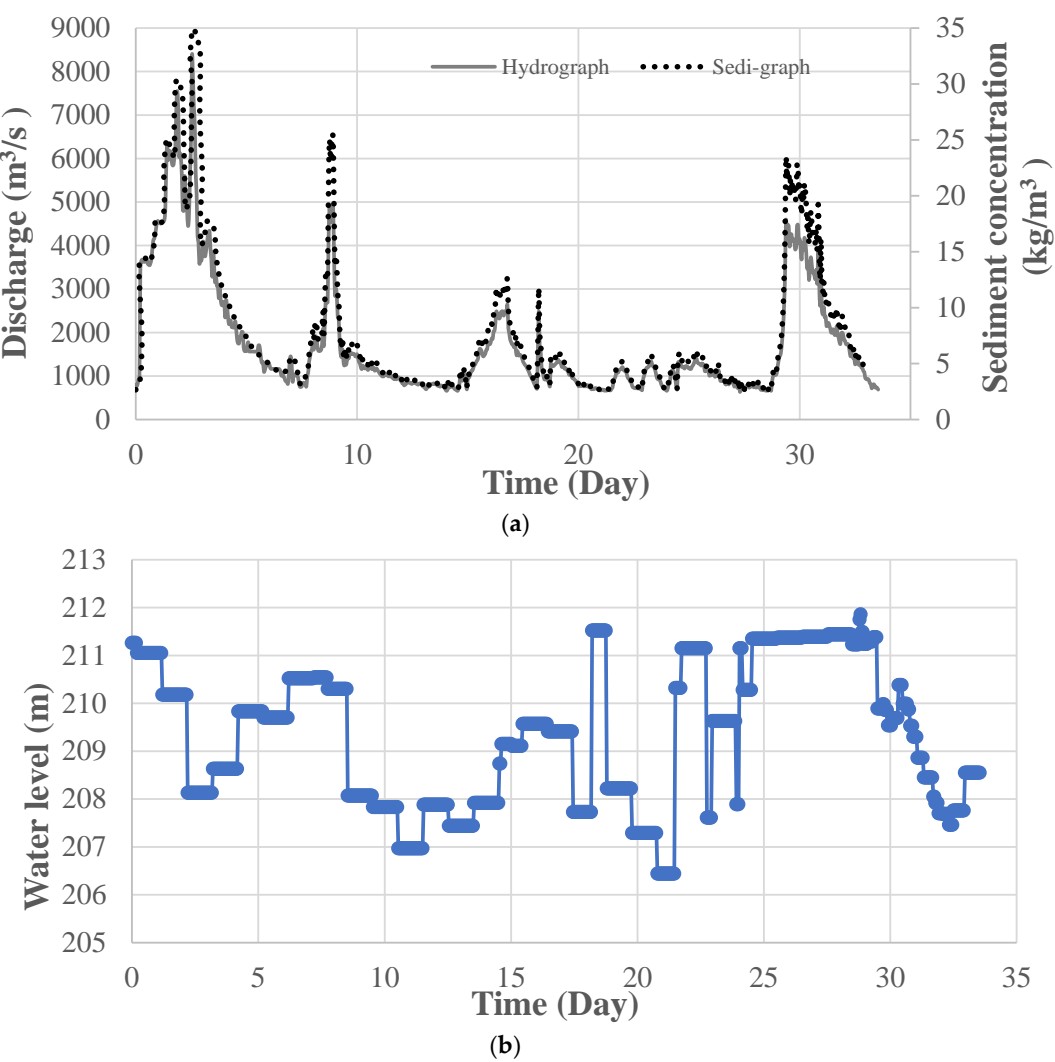

**Figure 14.** Boundary conditions (**a**) Hydrograph and sedigraph at JiLu Bridge. (**b**) Stage hydrograph at JiJi Weir.

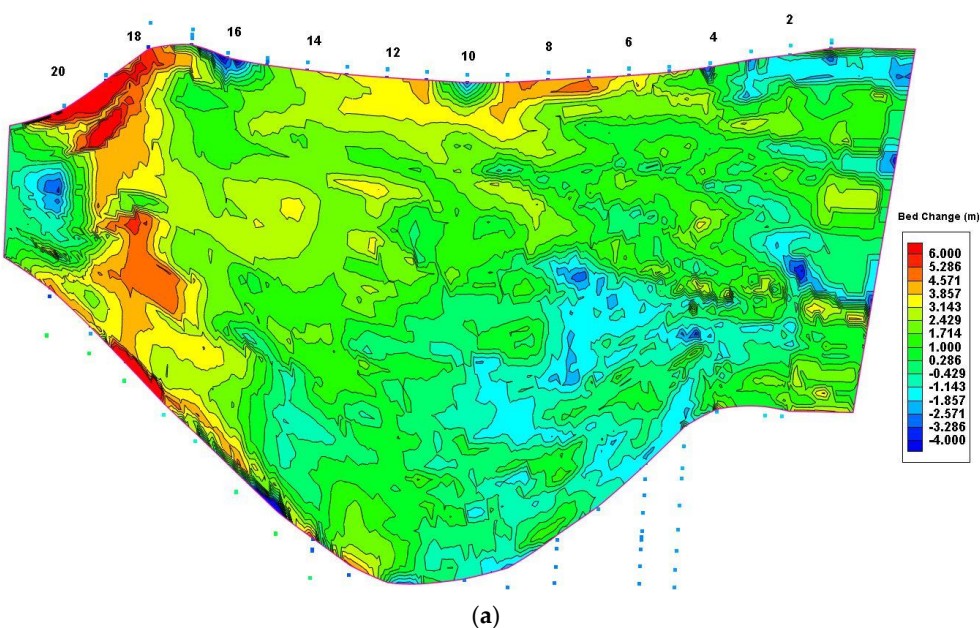

(**a**)

**Figure 15.** *Cont*.

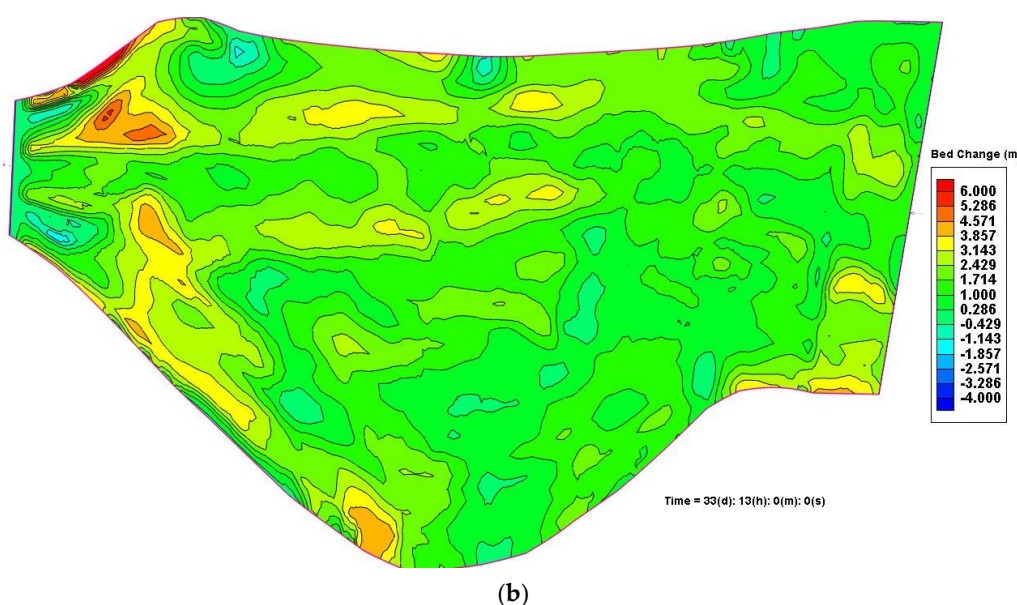

**(b)**

**Figure 15.** Bed changes from 2004 to 2007 in JiJi Reservoir. (**a**) Measured bed changes (**b**) Simulated bed changes.

Twenty measured cross sections were available within the JiJi Reservoir, and their locations are indicated in Figure 15a. Ten measured cross sections were compared with the simulated cross-section profiles, and the simulated bed-change patterns match them well (Figure 15b). As shown in Figure 16, from CS-2 to CS-12, the simulation predicted slight deposition on the left delta but captured the deposition on the right side. From CS-14 to CS-18, good agreement between the simulations and measurements was obtained. At CS-20, the simulation missed the scour hole in the middle of the cross section but predicted two smaller ones on two sides. Considering the uncertainties brought by the manual dredging activities on the left delta that were not considered in the simulation, the overall simulation results were reasonable.

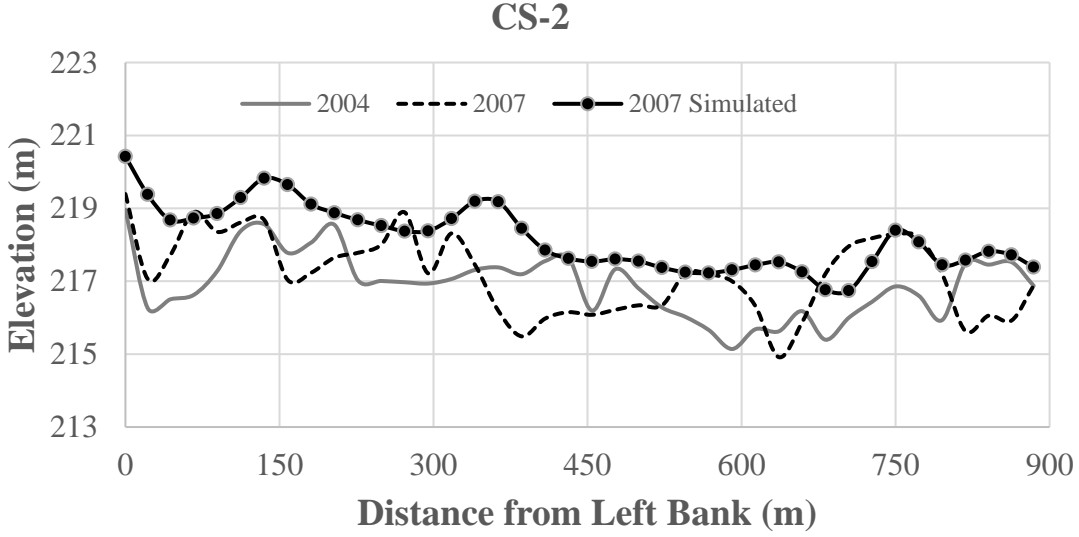

**Figure 16.** *Cont.*

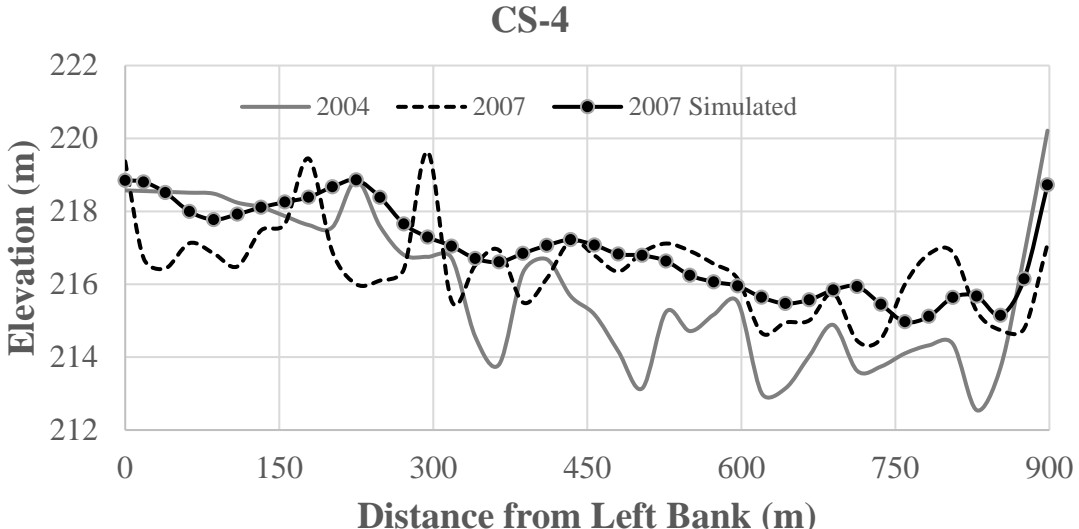

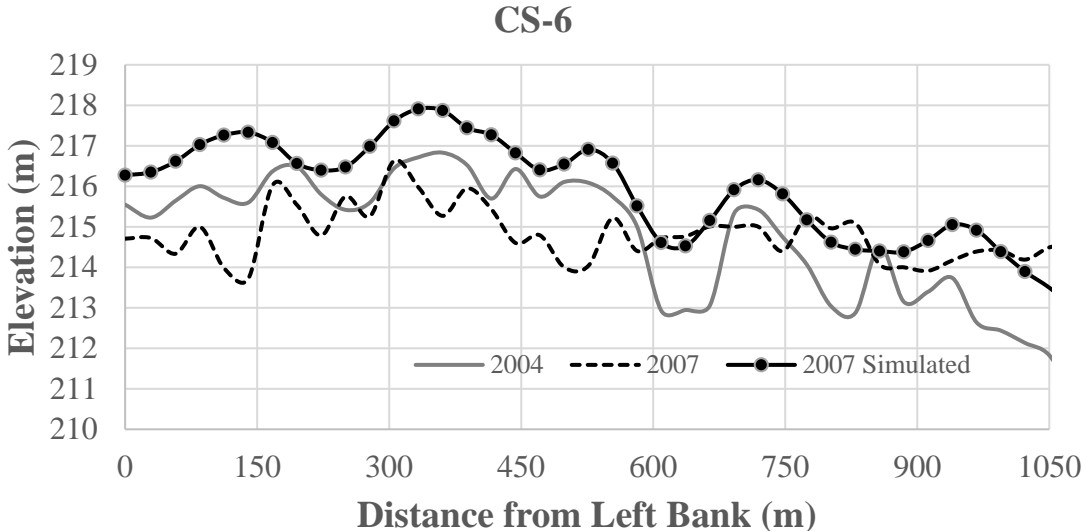

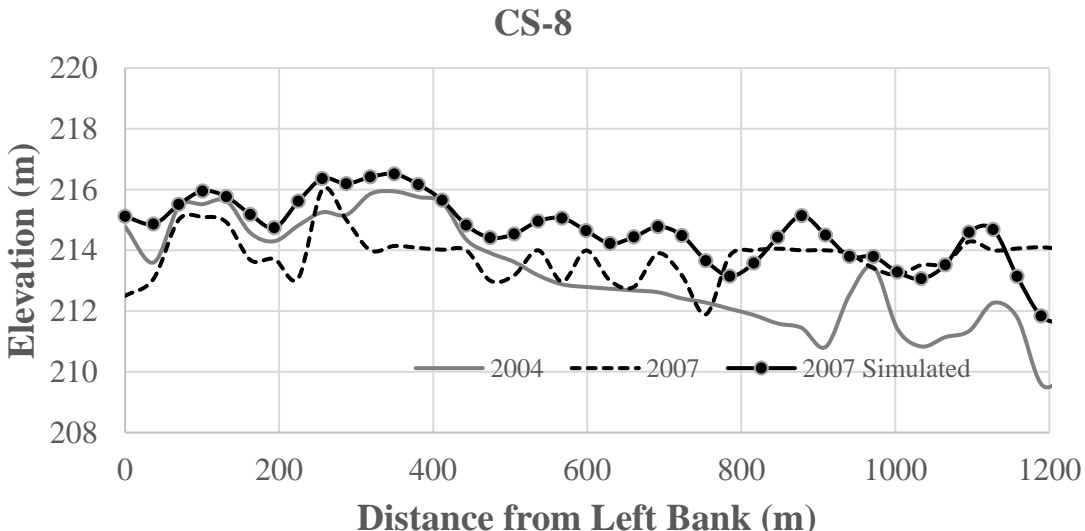

**Figure 16.** *Cont*.

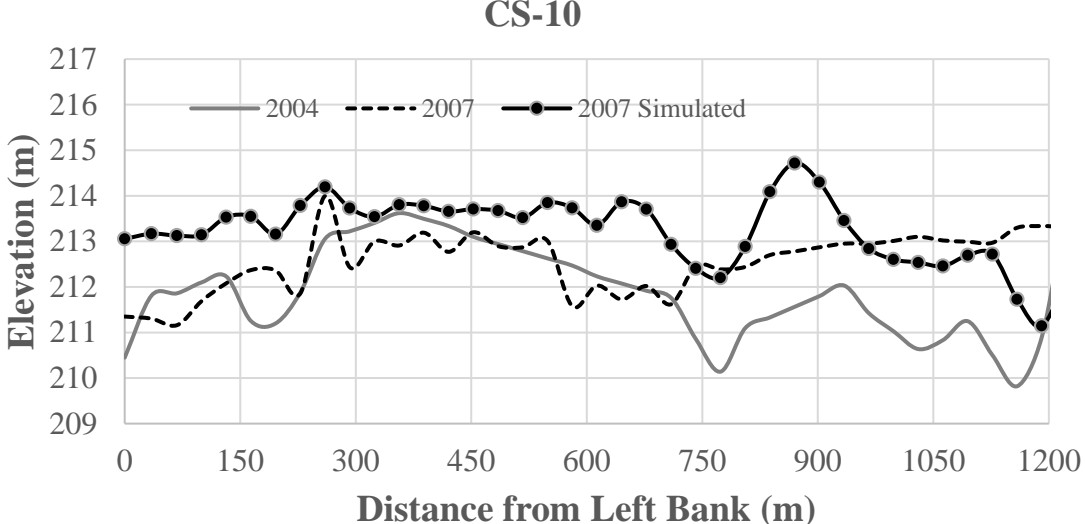

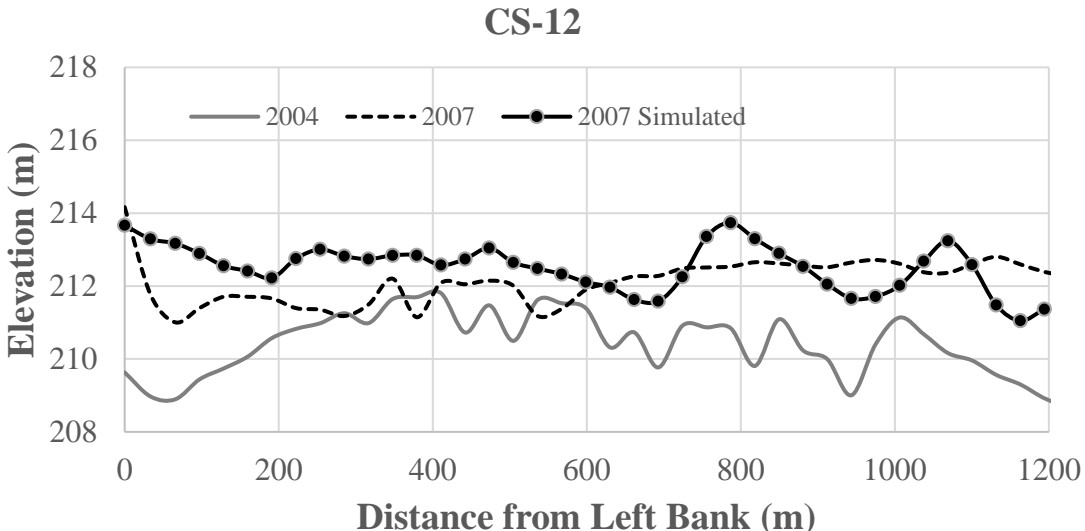

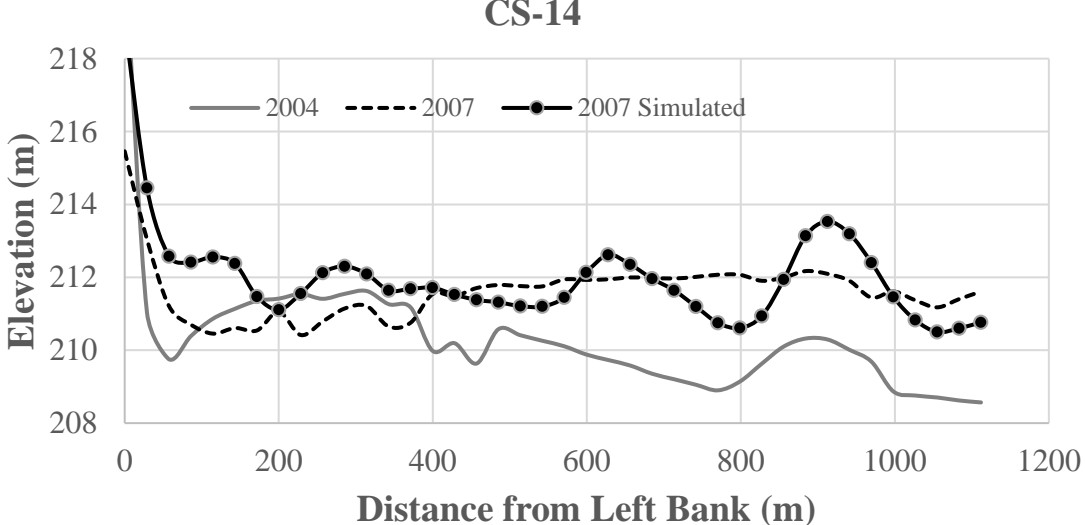

**Figure 16.** *Cont.*

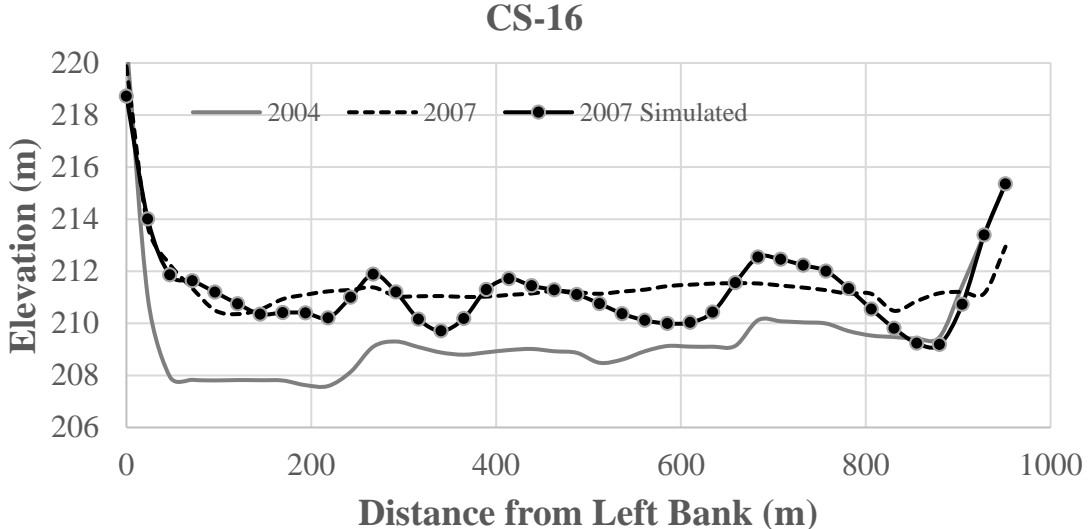

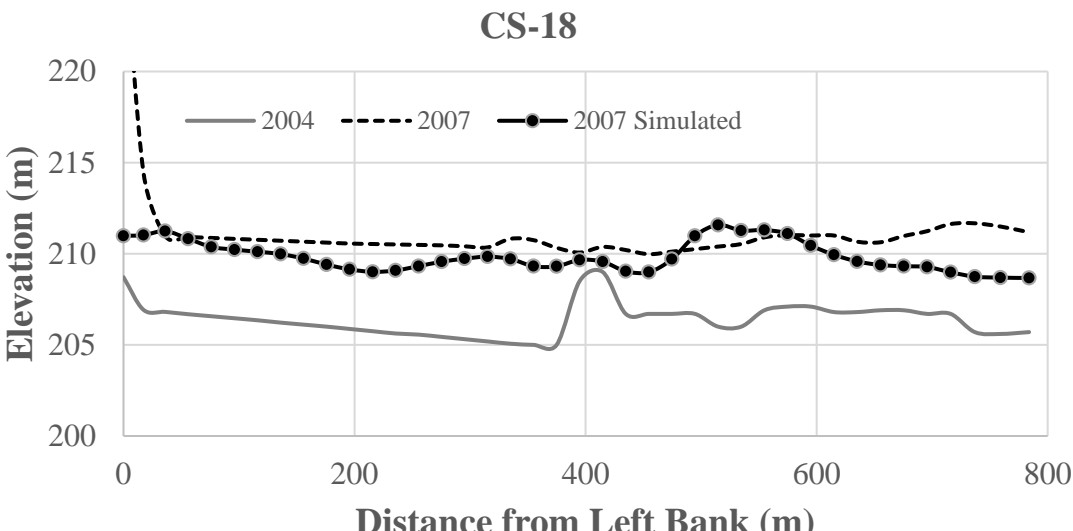

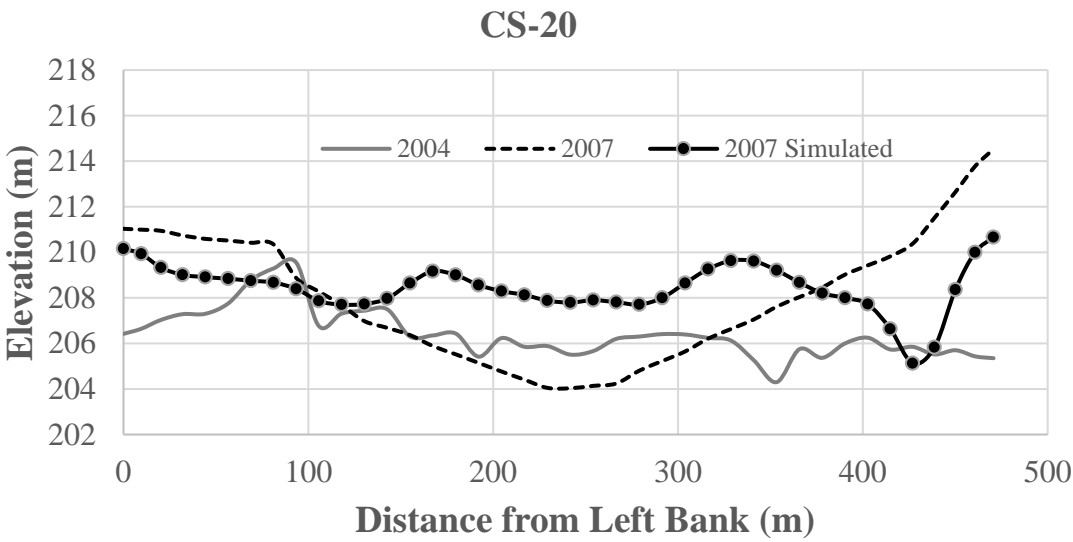

**Figure 16.** Comparisons of cross section profiles.

### 4.5. Discussion

Two laboratory-scaled cases and two field-scaled cases validated and demonstrated the developed sediment-transport model. For all cases, the fractional sediment transport played important roles, which was one of the model limitations identified by Papanicolaou et al. [34]. Multisized sediment particles may have interactions among each other in the transport or movement, such as collisions, flocculation, and disaggregation, which would affect the sediment-transport rates, concentrations, and settling behavior. Such interactions are better considered in two-phase flow models [21]. In this study, both the suspended-load and the bed-load transport-capacity formulas [41] considered the hiding-and-exposure effects in sediment mixtures, which partially and indirectly address the particle–particle interactions.

According to the comparisons with the measured data, the developed model obviously had better performances in laboratory cases than in field cases. One reason lies in the relatively simple and easy-to-control flow and sediment initial and boundary conditions in laboratory cases. In field cases, there are few well-monitored sediment data, and data scarcity has been a concern in sediment transport modeling. Specifically, in the case of the JiJi Reservoir, only limited bed samples were available, there was no inflow fractional sediment data, the measured cross section topography data were insufficient, and the dredging activities were not well recorded.

As stated previously, the flow model is the backbone of the sediment-transport model. Accurate flow calculations would provide solid conditions for qualitatively correct sediment-transport modeling. To ensure stable and accurate flow calculations, this study adopted second-order edge-gradient evaluation to consider corrections from mesh irregularities [36], and multipoint momentum-interpolation corrections for nonphysical oscillations in the wetting-and-drying process [51].

### 5. Conclusions

In this study, a 2D single-phase nonequilibrium sediment-transport model has been developed for nonuniform, noncohesive sediments in unsteady turbulent flows. The model was discretized using FVM on a hybrid unstructured-mesh system with high adaptivity for geometrically-complex domains. The conventional suspended-load transport equation and the bed-load transport equation were solved separately in the model. For the suspended-load gradient, the velocity gradient, and the pressure gradient at the edges, a second-order accurate method was used to take account of the mesh irregularities and avoid interpolations for the vertex values. For the possible nonphysical oscillations induced in the wetting-and-drying process, a multipoint momentum interpolation correction method was integrated to maintain both numerical stability and accuracy.

The sediment transport predictions were validated by a benchmark degradation case for the erosion process with armoring effects, a benchmark aggradation case for the deposition process, and a naturally meandering river was used for the validation of long-term unsteady sediment-transport processes. In general, the simulation results agreed well with the measured data, which indicates success in the development of this model. Finally, the model was successfully applied to simulate sediment transport in the JiJi Reservoir, which was significantly affected by typhoon events. In the future, this sediment-transport model will be improved and expanded in: (1) more applications for soil erosion and gully erosion in agricultural lands and (2) a cohesive sediment-transport process.

**Author Contributions:** Conceptualization, Y.Z.; methodology, Y.Z.; validation, Y.Z.; writing—original draft preparation, Y.Z.; writing—review and editing, D.W. and M.A.-H.; project administration, M.A.-H. All authors have read and agreed to the published version of the manuscript.

**Funding:** This work is supported by the U.S. Department of Agriculture, Agricultural Research Service, through the Cooperative Agreement No. 6060-13000-030-00D between the USDA-ARS National Sedimentation Laboratory (USDA-ARS-NSL) and the University of Mississippi/National Center for Computational Hydroscience and Engineering (NCCHE).

**Informed Consent Statement:** Not Applicable.

**Data Availability Statement:** All data in examples and application are available upon request.

**Acknowledgments:** The authors would like to thank Ron Bingner from National Sedimentation Laboratory (NSL), USDA Agricultural Research Service (ARS) for his suggestions and help in this study.

**Conflicts of Interest:** The authors declare that they have no known competing financial interests or personal relationships that could have appeared to influence the work reported in this paper.

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
