# Peer review of "Development of a Two-Dimensional Hybrid Sediment-Transport Model"

_applsci, doi:10.3390/app13084940_

Round 1

Reviewer 2 Report

·         The references order is hard to track you started with reference number 53, please reorder the reference in order.

·         Figure 2, there is only two legend(10m-13m) which the figure shows 6 legends. Same problem for figures (3-5)

Reviewer 3 Report

The manuscript develops a sediment transport model, and applies it to cases of bed degradation and aggradation, sediment transport, and reservoir siltation.   In general, it presents a current and scientifically relevant topic, the objective is clear, and the results are interesting. I recommend approval after minor revisions that I point out below:

(i) In the abstract, only the description of the model was presented, I suggest that the cases of applications of the model be included. I believe that it may increase the visibility of the manuscript, as some readers may be interested in the modeled processes and not the model.

(ii) Key words, I suggest that they be replaced so as not to repeat words present in the title.

(iii) In addition to the development of the model, examples of full-scale application are demonstrated, which simulate different hydrosedimentological processes. I suggest that this objective be explicit along with the others.

(iv) In the methodology it was not described how the sedimentological model performance was evaluated, uncertainty and predictive error values need to be demonstrated.

(v) Figures 2 and 3 can be grouped in a single figure a and b. The figure legends need to be improved, so that the reader can understand what is shown without recourse to the text. The same goes for figure 5.

(vi) Figure 7, did not bring any relevant information that justifies its presentation, and is shown in the work of Zhang et al (2015). I suggest deleting it.

(vii) Figures 8, 9 and 10 can also be grouped into a single figure. Standardize the numerical range and nomenclature of the X2 (Figure 8) and X1 (Figure 9) axes, water level or elevation? Add some metric to show numerically the fit of the model, for example R2, NS, mean error, or other. Report whether the results shown are from model fit or validation (data not used in the fit).

(viii) Figure 12 could be deleted, and the sample points shown in Figure 11 or 13. These two figures could also be merged into a single figure a and b. Add a scale in Figure 6 and 11. 

(ix) In Figure 14, add the legend inside the figure that the dotted line corresponds to sediment concentration.

(x) The elevation vs. flow relationship shown in figure 15, appears to be from a fitted model, data from natural systems hardly show this pattern. I suggest that this information be described in the legend, this figure could also be grouped with figure 14. 

(xi) Suggestion, a plot of elevation x area or elevation x sediment volume with the measured and simulated values, could show a general pattern about the Jiji Reservoir bed change. 

(xii) In the case of Jiji Reservoir, I also suggest adding some metric to numerically show the fit of the model, for example R2, NS, mean error or other.

(xiii) I missed a discussion that highlights the advances brought about by the model over others, and how these advances were reflected in improving the predictive performance of the model.

Reviewer 4 Report

Dear Authors. 

Thank you for your interesting work and study. The manuscript “Development of a Two-dimensional Hybrid Sediment 2 Transport Model” discusses 2D single-phase non-equilibrium sediment transport model development for non-uniform non-cohesive sediments in unsteady turbulent flows. Suspended and bedload transport integrated to model using empirical relationships. Calibration and model validation for different case studies show acceptable result for both suspended load and bedload simulation in open channel and natural streams.

The paper is generally well-structured and covers an interesting topic. However, the main body of the manuscript (methodology and models) needs several revisions. There are some question regarding the structure of the paper as well as some technical question regarding the paper which were mentioned inside the attached copy of the manuscript.

Overall, I recommend the paper for publication after addressing the comments/suggestions considering a Minor Revision.

·         Please see the attached PDF file of the manuscript with complete comments.

SB

Round 2

Reviewer 1 Report

The manuscript has been improved. Comments/Questions asked in the first review are addressed and can be considered for publication.

Reviewer 4 Report

Dear Authors, 

Thanks for your great work. The manuscript “Development of a Two-dimensional Hybrid Sediment Transport Model” developed a 2D single-phase non-equilibrium sediment transport model for non-uniform non-cohesive sediments in unsteady turbulent flows. The result are acceptable for various case studies and comments were addressed carefully. Thank you for considering the comments and your great work is appreciated. I have no other comments and I am confident that your work could be published with recent revisions you made.

Thank you!

Best